



# Assessing the long-term hydrologic response to wildfires in mountainous regions

Aaron Havel [1,2]; Ali Tasdighi [1,3]; Mazdak Arabi [1]

[1] Department of Civil and Environmental Engineering, Colorado State University, Fort Collins, CO 80523, USA
[2] AWR Engineering, LLC, Anchorage, AK 99519, USA
[3] Department of Civil and Environmental Engineering, University of California Irvine, Irvine, CA 92697, USA

*Correspondence to*: Ali Tasdighi (ali.tasdighi@colostate.com)

**Abstract.** This study aims to understand the long-term hydrologic responses to wildfires in mountainous regions at various spatial scales. The Soil and Water Assessment Tool (SWAT) was used to evaluate hydrologic response of the upper Cache la
Poudre watershed in Colorado to the 2012 High Park and Hewlett wildfire events. A baseline SWAT model was established to simulate the hydrology of the study area between the years 2000 and 2014. The effects of wildfires on land cover were accounted for in the model using the SWAT land use update module. The wildfire effects on curve numbers were determined comparing the probability distribution of curve numbers after calibrating the model for pre and post wildfire conditions. Daily calibration and testing of the model produced "very good" results. No-wildfire and wildfire scenarios were created and
compared to quantify changes in average annual total runoff volume, water budgets, and full streamflow statistics at different spatial scales. At the watershed scale, wildfire conditions showed little impact on the hydrologic responses. However, a runoff increase up to 75 percent was observed between the scenarios in sub-watersheds with high burn intensity. Generally, higher surface runoff and decreased subsurface flow were observed under post-wildfire conditions. Flow-duration curves developed for burned sub-basins using full streamflow statistics showed that less frequent streamflows become greater in
magnitude. A strong ($R^2 > 0.8$) and significant ($p < 0.001$) positive correlation was determined between runoff increase and percentage of burned area upstream. This study showed that the effects of wildfires on hydrology of a watershed are scale-dependent. Results also revealed that the wildfires had a higher effect on peak flows, which may increase the risk of flash floods in post-wildfire conditions. **Keywords:** Wildfire hydrologic impacts; Burn severity; Hydrologic modeling; Land use land cover change; SWAT; Curve number

## 1 Introduction

Assessing the hydrologic and water quality effects of wildfires is becoming more important as the frequency and severity of wildfires in the United States (U.S.) and other regions around the world have shown increasing trends (Westerling, 2016; Doerr and Santin, 2016; Ebel et al., 2012). Wildfires may have undesired consequences for water quality, carbon storage,



and ecosystem disturbance (Gould et al, 2016; Holden et al., 2012; Moody and Martin, 2009). Wildfire prone regions, often with increasing populations, are susceptible to loss of life and catastrophic destruction from floods and debris flows as a result of higher runoff and erosion under post-wildfire conditions (Moody et al., 2013). Assessing the hydrologic effects of wildfires in mountainous regions is particularly interesting as these areas often contain the headwater sub-watersheds

supplying water that is relied upon downstream (Viviroli et al., 2007). Wildfires could alter the timing and magnitude of runoff, subsurface flow, along with other hydrologic fluxes which eventually will reduce the reliability of water supply in these areas (Ebel et al., 2012).

Characterization of complex responses to wildfires is difficult due to the spatial variability of post-wildfire conditions (Moody et al., 2013). Wildfires can substantially change land use-land cover (LULC) and vegetation within watersheds,

which may subsequently result in altering hydrologic regimes including: (1) increased availability of rainfall for runoff by decreasing canopy interception (Moody and Martin, 2009; Robichaud et al., 2000), (2) increased base flow through the decrease of water normally lost through evapotranspiration (Neary et al., 2003), and (3) increased runoff velocities and reduced interception/storage through loss of ground cover, litter, duff, and debris (Moody and Martin, 2001). These alterations can cause increased hillslope erosion and may significantly alter terrestrial habitat. They may also increase

channel flooding, decrease channel stability, fill the streambed with fine sediment, and modify temperature regimes (Ryan et al., 2011).

Mathematical modeling is a useful and well accepted approach for improving our understanding of complex watershed processes (Kiesel et al., 2013). For example, watershed models have been used for simulating streamflow in mountainous regions to identify important hydrologic interactions and processes (Sanadhya et al., 2014). The Soil and Water Assessment

Tool (SWAT) has been used to characterize and quantify the effects of LULC change, climate change, and mitigation strategies on runoff, evapotranspiration, streamflow, groundwater and other hydrologic responses showing very good results in terms of model performance (Tasdighi et al., 2017; Motallebi et al., 2017; El-Khoury et al., 2015; Fan and Shibata, 2015; Foy et al., 2015). More specifically, numerous studies involving SWAT model development and calibration have been conducted to evaluate the hydrology in mountainous and snow-driven regions throughout the world, including this study

watershed (Foy et al., 2015); Cannonsville Reservoir watershed, New York (Tolson and Shoemaker, 2007); the Little River watershed, Tennessee (Zhu and Li, 2014); two Himalayan drainages of Nepal (Neupane et al., 2015); and the Yingluoxia watershed of northwest China (Lu et al., 2015). These studies document promising results for application of SWAT for hydrologic simulations in mountainous regions.

One of the challenges in using models for evaluating the hydrologic response of a system to wildfire is developing the

mechanism through which the hydrologic effects of wildfires are simulated. Studies have used alteration of model parameters and LULC to represent pre and post wildfire conditions (Parson et al., 2010; Robichaud, 2000). The majority of these studies have used field measurements or implemented a deterministic approach using fixed values for specific



parameters during the pre and post wildfire conditions to represent the change in hydrology as a result of wildfire (Parson et al., 2010; Robichaud, 2007). This approach has a direct impact on the results obtained as selection of parameters to change and the values assigned to them during pre and post wildfire conditions are subjective and may not necessarily represent the changes in the real world. Additionally, lack of components for representing LULC change adds to the complexity of the

procedure (Batelis and Nalbantis, 2014; Goodrich et al., 2005; McLin et al., 2001). A proper LULC change component is required for continuous simulation and is particularly important when assessing effects of wildfires. The LULC change module within SWAT has been shown useful for evaluating hydrologic condition where LULC has changed as the result of urbanization (Pai and Saraswat, 2001).

The 2012 Hewlett and High Park wildfires have provided a unique opportunity for examining hydrologic response to

wildfires, specifically, in a mountainous region. This unique opportunity stems from the fact that a relatively significant proportion of the gaged Cache la Poudre (Poudre) headwaters (approximately 14 percent from the Mouth of Canyon) has been burned as the result of wildfire. The pre and post-wildfire streamflow data availability allows for the development, calibration, and testing of a hydrologic model that accounts for spatial variability in LULC to continuously simulate the hydrology from pre-wildfire conditions through post-wildfire conditions. Due to the magnitude of the 2012 wildfire incident,

burn severity mapping is available for the area. This mapping data allows for a land use change module to be implemented during calibration efforts which adjusts hydrologic parameters impacted by wildfire seamlessly during simulation.

The overall goal of this study is to characterize and quantify long-term hydrologic responses to wildfires in mountainous regions at various spatial scales (smaller high burn intensity sub-watersheds to watershed scale). To accomplish this goal, the Poudre headwaters located in northern Colorado, USA which experienced the 2012 Hewlett and High Park wildfires was

analyzed with the SWAT model. This analysis includes simulation of no-wildfire and wildfire scenarios over a 15 year (2000 to 2014) period. Specific objectives of this study are to: (1) quantify changes in average annual total runoff volume and explore how these changes fluctuate with the percent of the area burned, (2) quantify annual changes in various components of hydrologic budget, and (3) highlight potential implications of these changes using full streamflow statistics through application of flow duration curves at both sub-watershed and watershed scales. While a number of previous studies have

examined the hydrologic effects of wildfires, application of a probabilistic approach for characterizing the change in key hydrologic parameters between the pre and post wildfire scenarios and a dynamic LULC updating through the analysis period is novel. The results of this study have important implications for hydrologic effects of wildfires and methods used to assess them.



## 2 Material and methods

### 2.1 Study area

The Poudre Watershed, with an area of approximately 5,230 km2 above its confluence with the South Platte River on the Great Plains, is situated mostly in northern Colorado, USA with a portion reaching into southern Wyoming, USA (Wohl,
2010). The Poudre River (Figure 1) is supplied by two major tributaries within its headwaters, the South and North Forks, the latter being the longer of the two joining the main-stem farther downstream. After streamflow retreats from the Poudre's headwaters in the Rocky Mountain Range, the river passes through the cities of Fort Collins and Greeley. Eventually, the river joins the South Platte River and winds downstream to join the Platte River and then to the Missouri River. The Poudre River, with its minimally-developed mountainous headwaters, is widely utilized as a source of drinking water for several
cities and communities located along its banks (Richer, 2009).

During May and June of 2012 the Hewlett and High Park wildfires burned approximately 384 km2 of primarily forested landscape within the Poudre Watershed. The burned area includes numerous drainage tributaries to the main-stem of Poudre River. Widespread loss of vegetation and burned soils from the wildfires created areas susceptible to severe erosion and flooding. Localized summertime thunderstorms immediately following the wildfire worsened the effects by washing
sediment and debris into the river channel posing a threat to the safety of people and homes in the area (Oropeza and Heath, 2013). The affected area extends along the Poudre River from the mountain front upstream to several kilometers south of the community of Rustic, Colorado. Therefore, a study watershed outlet was defined near the mountain front at Colorado Division of Water Recourses' (CDWR) surface water gauge CLAFTCCO18 (formally USGS Gage 06752000), commonly referred to as the Mouth of Canyon (Figure 1).

The resulting study watershed is approximately 2,732 km2. At higher elevations, streamflow is dominated by snowmelt runoff and at lower elevations rainfall runoff from summer convective storms greatly affect streamflow. The storms combined with the upstream snowmelt runoff, can produce high-magnitude, short-lived floods at times (Wohl, 2010). The resulting hydrograph is snowmelt dominated with a rise typically beginning in April and a recession lasting into August. Generally, peak streamflow occurs at the end of May or early June and base flow levels occur in September or October
(Richer, 2009).

### 2.2 Hydrologic model

#### 2.2.1 SWAT model

SWAT is a continuous-time, distributed-parameter, process-based watershed model, which has been used extensively for hydrologic and water quality assessments under varying climatic, land use, and management conditions in small watersheds
to large river basins (Gassman et al., 2007; Neitsch et al., 2011; Arnold et al., 2012, CARD Staff, 2016).



SWAT model allows for numerous physical processes to be simulated in a watershed. These processes may be separated into two coarse divisions of the hydrologic cycle: the land phase and the routing phase. The main processes include precipitation, infiltration, surface runoff, evapotranspiration, groundwater flow, snowmelt, and flood routing. SWAT is driven by a water balance equation which relates individual components of the hydrologic cycle. Additional details including specific

equations associated with the water balance and the individual hydrologic processes may be found in the SWAT Theoretical Documentation, Version 2009 (Neitsch et al., 2011).

Compared to event-based models, continuous time models better represent watersheds where channel storage may be significant and/or where significant variability exists in land use (e.g., urbanization), soil types, and/or topography (Nicklow et al., 2006). Being a distributed-parameter model SWAT divides a watershed into sub-basins, which are further divided into

hydrologic response units (HRUs). HRUs are the smallest spatial units in SWAT, and are defined as areas within each subwatershed with unique combinations of land use, soil, and slope class. Sub-basins can be assigned unique climate and hydrologic properties which in combination with unique land use characteristics of HRUs provides the capability to investigate the effects of land use change scenarios under varying climatic conditions both spatially and temporally.

### 2.2.2 Updating LULC

The hydrologic modeling process was initiated by first collecting and preparing the necessary data, summarized in Table 1. A detailed description of each data type is presented in Appendix A.1. The NLCD 2011 Land Cover spatial dataset was preprocessed to allow the High Park and Hewlett wildfires to be simulated by SWAT. The NLCD 2011 Land Cover was overlaid with the Thematic Burn Severity dataset (Monitoring Trends in Burn Severity Project, 2014). Then the NLCD 2011 Land Cover was reclassified to incorporate low, medium, and high burn severity categories. The reclassification was

accomplished using spatial analyst toolset within ArcMap from ESRI's ArcGIS software package. The preprocessing retained the pre-wildfire classification, but added a burn severity identifier. For example, portions of the NLCD 2011 Land Cover that consist of Evergreen Forest and overlap with a low burn area were reclassified to a newly created Evergreen Forest Low Burn classification.

The SWAT Model Database contains various pre-defined model parameters for different LULC types and the SWAT LULC

lookup table relates NLCD classifications to the LULC types found in the SWAT Model Database. In order to seamlessly represent wildfire during the simulation the SWAT Model Database and SWAT LULC lookup table were altered to reflect the conditions after the wildfire.

For the pre-wildfire database, the newly added LULC types consisted of attributes identical to the original classification, but with a new description and identification code. Thus, the SWAT model created using this database will represent pre-

wildfire condition, but areas influenced by wildfire will be delineated from non-burned areas.



For the post-wildfire database, the newly added LULC types included a new description and identification code similar to the pre-wildfire database; however, for the post-wildfire case, attributes were also altered from their original classifications. For all burned areas, LULC attributes in the database were changed to match those of the Range-Grasses LULC. This change was implemented to aid with appropriately representing loss of canopy in burned areas.

### 2.2.3 Updating Curve Numbers

Curve Numbers (CNs) were adjusted to account for expected increases in runoff. The change in CNs was based on a pre and post wildfire calibration of the model explained in section 2.2.7. Comparing the probability distribution of CNs before and after the wildfire, an average CN increase of 5, 10, and 15 for areas with low, moderate, and high burn intensity were considered respectively. The original and edited SWAT LULC lookup tables as well as curve numbers for both pre and post-wildfire conditions can be found in Appendix A.2, tables A3 through A5.

### 2.2.4 Initial model development

Two models representing pre and post-wildfire conditions were developed and then later merged to create a unified model. Two sets of initial SWAT model input files for the study watershed were created using ArcSWAT 2012 (U.S. Department of Agriculture Agricultural Research Service, 2014). ArcSWAT is an ArcMap extension that provides a graphical user interface for creating a SWAT model. The interface was used to process the previously described model data to generate initial SWAT input files. This process is summarized in Figure 2.

The ArcSWAT Automatic Watershed Delineation tool was used to create a stream network, define sub-basin outlet locations, delineate the watershed, and calculate the sub-basin parameters. Additional outlets were manually placed at locations where a large tributary entered the study reach and the whole watershed outlet was defined at the Mouth of Canyon. ArcSWAT was then used to determine LULC/soil/slope combinations within each sub-basin which is then used to determine the distribution of HRUs for the entire watershed.

### 2.2.5 Model options

Options for both models are identical and were selected based on previous modeling studies using SWAT in mountainous regions (Foy et al., 2015; Lu et al., 2015; Neupane et al., 2015). A modified version of the commonly applied United States Soil Conservation Service (now the NRCS) curve number (CN) procedure was adopted to simulate surface runoff in the watershed. The CN depends on the soil type, LULC, and hydrologic condition (Lu et al., 2015). Penman-Monteith method based on energy balance components was selected to estimate potential evapotranspiration. Lastly, channel routing was represented using the Muskingum River Routing Method. Other model options were left as default configurations.



### 2.2.6 Accounting for Mountainous Terrain

The study watershed is located within the rainshadow of the Rocky Mountains and overall experiences a topographically-driven climate. Significant difference in elevation within the study watershed yields large variability in the quantity and form of precipitation. Thus, lapse rates as well as elevation band parameters were assigned to each sub-basin to account for

orographic effects. The precipitation lapse rate (i.e., increase in mean annual precipitation with an increase in elevation) of 658.4 mm/km obtained from Foy (2015) was incorporated into the model. Additionally, the temperature lapse rate (i.e., decrease in mean annual temperature with an increase in elevation) of -5.5 $^{0}$C/km reported by Foy (2015) was used.

SWAT is capable of integrating up to 10 elevation bands in each sub-basin. These bands were derived by topographically discretizing each sub-basin within the watershed. SWAT requires the input of the elevation at the center of each band and the

10 fraction of sub-basin area within the elevation band. Data from the Topography Report generated by AcrSWAT was used to discretize each sub-basin into 10 elevation bands. The minimum elevation was subtracted from the maximum elevation and divided by ten, which creates ten equal-interval elevation bands. Next, the elevation at the center of each band and the fraction of sub-basin area within the elevation band is calculated. Lastly, the previously generated SWAT input files were modified to contain these parameters. These parameters allow SWAT to use the elevation band equations described in the

15 SWAT Theoretical Documentation, Version 2009 (Neitsch et al., 2011) to simulate orographic effects.

The curve numbers provided in the SWAT Model Database are appropriate for slopes up to 5 percent (Neitsch et al., 2011). An analysis of the HRUs revealed that many of the average HRU slopes in the study watershed exceed 5 percent. We adjusted the curve numbers for different slopes at the HRU level using the following equation (Neitsch et al., 2011):

$$CN_{2s} = \frac{(CN_3 - CN_2)}{3} * \left[1 - 2 * e^{-13.86*S}\right] + CN_2 \,, \tag{1}$$

where $CN_{2S}$ is the moisture condition II CN adjusted for slope, $CN_3$ is the moisture condition III CN for the default 5 percent slope, CN2 is the moisture condition II CN for the default 5 percent slope, and S is the average fraction slope of the sub-basin. Note that upon simulation SWAT caps CN values at 98.

### 2.2.7 Land use update module

The pre and post wildfire models were used to create a unified model by activating the land use update module in SWAT. A

25 Matlab code was developed to prepare the land use update files. This code was prepared to add the burned HRUs from the post-wildfire model to the pre-wildfire model and create a land use update file to make a unified model. Land use update files tell SWAT to change the pre-wildfire HRU fractions to nearly zero (The HRU fraction is a HRU level parameter that specifies the fraction of sub-basin area represented by that HRU, Neitsch et al., 2011) and increase the post-wildfire HRU fractions to represent the burn area at the appropriate time during the simulation. In this case, the High Park and Hewlett





wildfires occurred during May and June of 2012. Thus, for model calibration the land use update was initiated on July 1, 2012.

### 2.2.8 Model calibration and testing

The SWAT model was calibrated and tested for the daily naturalized streamflows at the Mouth of Canyon. The naturalized flows were determined using the daily historical flow records of diversions or reservoirs. Separate calibrations were conducted for pre and post wildfire conditions with the purpose of characterizing the change in CN as a result of wildfire. Once the change in CN was characterized, it was implemented in the unified model. The unified model was then calibrated and tested for the whole period (2000-2014). Calibration, pre-wildfire testing, and post-wildfire testing periods were 2005-2013, 2000-2004, and 2014, respectively. These simulation periods were selected based on data availability. Initial calibration parameters were identified from previous modeling efforts for the study watershed published in Foy (2015). These parameters were supplemented with additional parameters identified from a previous sensitivity analysis study utilizing SWAT (Ahmadi et al., 2014). A total of 38 modal parameters were used for calibration. A Matlab code was developed for auto-calibration of SWAT model using a global optimization algorithm named dynamically dimensioned search (DDS; Tolson and Shoemaker, 2007). DDS is designed to arrive at good solutions within a maximum number of user-defined function evaluations for use in model calibration with many parameters (Tolson and Shoemaker, 2007). This auto-calibration tool was used to generate 498 model runs. Each model run consisted of a unique combination of the 38 model calibration parameters. The tool works towards minimizing an objective function. In this case, we based this objective function on two primary error statistics, relative error (RE) and Nash-Sutcliffe efficiency coefficient ($E_{NS}$). $E_{NS}$ is a normalized statics that indicates how well observed versus simulated plot fits a 1:1 line. $E_{NS}$ is computed as:

$$E_{NS} = 1 - \left[ \frac{\sum_{i=1}^{n}\left(Y_i^{obs} - Y_i^{sim}\right)^2}{\sum_{i=1}^{n}\left(Y_i^{obs} - Y_i^{mean}\right)^2} \right], \tag{2}$$

where for this study $Y^{obs}$ is the observed streamflow, $Y^{sim}$ is the simulated streamflow, and $Y^{mean}$ is the mean of observed streamflows (Nash and Sutcliffe, 1970). The optimal value for $E_{NS}$ is 1. $E_{NS}$ can range between -∞ and 1.0, with values between 0.0 and 1.0 generally regarded as satisfactory levels of performance. Values equal to or smaller than 0.0 indicate that the simulated values are a same or worse predictor than the mean of observed values (Moriasi et al., 2007) respectively. The RE gives an indication of how good simulated values are relative to the magnitude of corresponding observed values. RE in percentage is computed as:

$$RE = \frac{\sum_{i=1}^{n}\left(Y_i^{obs} - Y_i^{sim}\right)}{\sum_{i=1}^{n}Y_i^{obs}} * 100, \tag{3}$$





where for this study $Y^{obs}$ is the observed streamflow and $Y^{sim}$ is the simulated streamflow. These error statistics are used to determine how accurately SWAT is representing hydrologic processes through comparison of observed and simulated streamflows at the Mouth of Canyon. Model calibration parameter starting values and ranges are displayed in Appendix A2, table A6.

### 2.2.9 Scenario analysis

With the SWAT model calibrated and tested, two scenario models were created. First, a no-wildfire scenario model was created. This was achieved by simply removing the land use update files thus representing no wildfire activity throughout the entire simulation period. Second, a wildfire scenario model was created. This was achieved by adjusting the land use update files to reflect a wildfire occurring at the beginning of the simulation. Thus, wildfire is simulated throughout the entire simulation period. Note the simulation period for each scenario was between 2000 and 2014 (15 years).

### 2.3 Output Data Post-Processing

SWAT outputs were post-processed in Matlab. Simple summing functions were used to calculate total runoff volumes and water budgets throughout the study watershed. Full streamflow statistics were used to develop flow-duration curves for burned sub-basins. These represent the percentage of time that streamflow is likely to equal or exceed a given streamflow value for both scenarios. The code used sorts, ranks, and plots the input streamflow data to generate flow-duration curves. Flow-duration curves are a widely accepted method for characterizing streamflow regime. They are commonly used for hydropower, water resource management, water quality management, habitat suitability, and flood control applications (Fan and Li, 2004). However, they have not been frequently used in evaluating response to wildfires (Newtson, 2013). Next, the ecodeficit and ecosurplus metrics introduced by Vogel (2007) were computed for each flow-duration curve. These metrics provide a simplified representation of hydrologic impacts (Vogel et al., 2007). For this study, ecodeficit is defined as the ratio of the area below the no-wildfire scenario flow-duration curve and above the wildfire scenario flow-duration curve divided by the total area under the no-wildfire scenario flow-duration curve. Conversely, ecosurplus is defined as the ratio of the area above the no-wildfire scenario flow-duration curve and below the wildfire scenario flow-duration curve divided by the total area under the no-wildfire scenario flow-duration. Thus, these values represent the overall loss (ecodeficit) and gain (ecosurplus) in streamflow (Vogel et al., 2007) between scenarios.

## 3 Results and discussion

### 3.1 Model performance

The performance of the model during calibration and testing at various temporal scales was assessed using the common criteria in the literature (Motovilov et al., 1999; Moriasi et al., 2007). Model performance at a daily timestep was considered





good if $E_{NS} \geq 0.75$ and was considered satisfactory for values of $E_{NS}$ between 0.75 and 0.36 (Motovilov et al., 1999). At monthly timestep, the performance of the was categorized as very good ($0.75 < E_{NS} \leq 1.00$), good ($0.65 < E_{NS} \leq 0.75$), satisfactory ($0.5 < E_{NS} \leq 0.65$), and unsatisfactory ($E_{NS} \leq 0.5$) (Moriasi et al., 2007).

During the separate pre and post wildfire calibration the model had a good performance giving $E_{NS} = 0.92$; RE = -0.32%, and

$E_{NS} = 0.81$; RE = 2.16% respectively. This separate calibration step was done to characterize the change in CN between pre and post wildfire conditions. Figure 3 illustrates boxplots of CNs for the pre and post wildfire conditions. The values of CN for the best solution (highest $E_{NS}$) and mean of the CNs are also marked on the boxplots. Based on these results, an average CN increase of 5, 10, and 15 for areas with low, moderate, and high burn intensity were considered respectively.

The optimal parameter set found during the calibration effort generally yielded good results. The model performed best

during the post-wildfire testing period, but still performed well during the calibration period and pre-wildfire testing period. Final values for the 38 calibration parameters are displayed in Appendix A2, table A6. Model performance was evaluated based on primary statistical results (at both the daily and monthly timesteps) and visual inspection of the graphical results.

The best calibration achieved for the Mouth of Canyon naturalized streamflow at the daily timestep was $E_{NS}$ of 0.82 and RE of 1.68. The testing $E_{NS}$ values for the pre-wildfire and post-wildfire periods were 0.71 and 0.88, with RE values of -19.52%

and 9.31%, respectively. Table 2 presents a summary of the model performance at the daily timestep.

All simulation periods earned a performance rating of very good at the monthly timestep. Monthly results were generally comparable to those from other SWAT modeling studies involving mountainous watersheds (Foy et al., 2015; Lu et al., 2015; Neupane et al., 2015). Table 3 presents a summary of the model performance at the monthly timestep.

Generally, simulations yielded good visual agreement between observed and simulated daily streamflows and total runoff

volume, as shown in Figure 4. A slight discrepancy between the observed and simulated total runoff volume exists for the no-wildfire testing period. This difference propagates to the statistical results, most notably, the RE value of -19.52%. A negative relative error shows that the model overestimates runoff volume compared to observations. Based on visual examination of the hydrographs, the calibration period may be slightly "wetter" relative to the pre-wildfire testing period, which may be the cause of the noted discrepancy.

Also, the simulated and observed flow-duration curves for the entire simulation period yielded good visual agreement, as shown in Figure 5. The simulated flow-duration curve generally follows the observed flow-duration curve with the exception of a slight deviation for less frequent flows. For the less frequent streamflows the model is underestimating streamflows. A deviation is expected as less frequent streamflows correspond to larger streamflows which are less predictable and less understood.

Previous studies have used SWAT along with similar calibration techniques throughout this region for hydrologic analysis. However, use of the SWAT land use change module to investigate hydrologic response to wildfires has not been well documented. Moreover, characterization of change in CNs as a result of wildfires using a probabilistic approach has not been



performed previously. The performance results above indicate that the comprehensive methodology of using the SWAT land use change module along with multi-variable parameter calibration was an effective technique to represent the hydrology of an area which has been exposed to wildfire

## 3.2 Model performance

The daily simulation outputs from both no-wildfire and wildfire scenarios were analyzed and compared in order to characterize an average hydrologic response to wildfire during the simulation period of 15 years (2000 to 2014). Total runoff values, represented as both depth and volume for each burned sub-basin as well as for the entire study watershed are shown in Table 4. Also, Figure 6 displays the burn severity distribution and average annual total runoff percent increase (based on the values presented in Table 4) for each burned sub-basin and for the entire study watershed. The average annual total
runoff includes surface runoff, lateral flow, and base flow.

Figure 6 shows that in the case of sub-basins 28, 30, 26, and 32, more than 50 percent of the area experienced burning as a result of the High Park and Hewlett wildfires. Sub-basins 28 and 30 were the most severely burned with large high burn severity percentages. The remaining sub-basins had smaller burned area percentages.

The total runoff percent increase between scenarios was greatest on average for sub-basins 28, 30, 26, and 32. For these sub-
basins, increases in runoff between the no-wildfire and wildfire scenarios ranged from approximately 66 to 75 percent. For the remaining sub-basins, as well as the entire study watershed, runoff percent increases are found to be considerably less. This is likely because those sub-basins were not as heavily burned. Nevertheless, the results indicate wildfire effects at larger scales are still substantial, but only in terms of the magnitude rather than percent change of total runoff volume increase. Larger areas (i.e., sub-basin 35 and the entire study watershed) appear to experience much greater absolute increases in total
runoff volume between scenarios, despite having smaller total burn area percentages. This is what we might expect given that each sub-basin is nested within the study watershed, resulting in a cumulative effect.

Other studies have documented total runoff increases under post-wildfire conditions (Benavides-Solorio and MacDonald, 2001; Inbar et al., 1998; Lavabre et al., 1993; Robichaud et al., 2000; Scott, 1993). For example, Lavabre et al. (1993) used a lumped conceptual hydrological model to evaluate a small Mediterranean basin which experienced a burn covering 85
25   percent of its surface area in 1990. They suggested a 30 percent increase in the annual runoff yield. Scott (1993) showed total streamflow volume increases of 15.3 and 9.4 percent in response to burning in two small mountainous catchments using a paired catchment method. In contrast, Mahat et al (2015) reported no significant change between the modeled streamflow from burned and unburned models. They suggested that this outcome may be the result of using a conceptual modeling approach instead of using a physically based model. The amount of total runoff volume increase following wildfire
disturbance varies greatly between locations depending on wildfire intensity, proportion of the forest vegetation burned, climate, precipitation, geology, soils, watershed aspect, and tree species (Neary et al., 2003). Thus, it is not surprising that



results vary. Also, comparison between studies is difficult because of changes in size of disturbance (i.e., wildfire) in relation to the size of the catchment (Robichaud et al., 2000). This emphasizes the need to examine increases based on percent burn area upstream.

Figure 6 is arranged in descending order of percent burned area from left to right. Generally, we see an increase in total runoff as percentage of total burn area increases. This observation is consistent with reports in the literature indicating total runoff volume increase following wildfire disturbance is in part a function of the proportion of the contributing area burned (Neary et al., 2003; Robichaud et al., 2000). This relationship is further explored by applying linear regression to the data. Figure 7 shows a linear regression model fitted between the total runoff volume increase and total burned area percentage. Note that the entire study watershed results were not included in this regression. Also, sub-basin average slope was categorized as low (slope < 0.30), moderate ($0.30 \leq$ slope < 0.40), and steep (slope $\geq$ 0.40) for each sub-basin.

An F-test was performed using Matlab to determine if this particular model fits the data well. The regression generally yields a good fit, with a p-value < 0.001 for the F-test. No previous study was found documenting this relationship with linear regression. Thus, this study suggests it may be reasonable to use total burn area percentage as a predictor for increase in total runoff volume. Also, the figure indicates that generally for the High Park and Hewlett wildfires the sub-basins with moderate to steep slopes experienced wildfire in a larger percentage of their area relative to low slope sub-basins.

### 3.3 Wildfire effects on hydrologic budgets

The daily simulation outputs from both no-wildfire and wildfire scenarios were further analyzed and compared in order to quantify changes in average annual hydrologic budgets as a result of wildfire during the simulation period of 15 years (2000 to 2014). Figure 8 shows hydrologic budgets for select sub-basins as well as the entire study watershed. These hydrologic budgets show the fate of average annual precipitation along with the fate of average annual total runoff. The fate of precipitation (rainfall and snowfall) is shown as evapotranspiration, total runoff, and other (deep aquifer contribution and soil water storage). Also, the major hydrologic processes for the fate of runoff were defined as surface and subsurface (lateral flow and base flow) runoff.

It is evident that hydrologic budgets change on the sub-basin scale following wildfire; however, little change is seen at the watershed scale. Batelis and Nalbantis (2014) also documented that wildfire effects are practically indiscernible on a regional scale. Generally, Figure 8 shows under the wildfire scenario an increase in surface runoff and a corresponding decrease in subsurface flow at the sub-basin scale. For example, the hydrologic budget for sub-basin 30 (a heavily burned area) shows a change in surface runoff from 21 to 61 percent under the no-wildfire and wildfire scenarios, respectively. This is consistent with previous studies in which it seems to be generally accepted that infiltration rates decrease after wildfires. For example, Moody and Martin (2001) showed that infiltration rates were decreased by a factor of two to seven after wildfires.



At the sub-basin scale under the wildfire scenario we also see less evapotranspiration. This connects well with the results from Section 3.2, where generally we see an increase in total runoff for the wildfire scenario. Increased water yields (i.e., total runoff) primarily due to reduced evapotranspiration has been a reported effect on post-wildfire hydrology (Neary et al., 2003; Townsend and Douglas, 2004).

## 3.4 Implications of wildfire effects

Lastly, the daily simulation outputs from both no-wildfire and wildfire scenarios were analyzed and compared in order to determine potential implications of wildfire effects during the simulation period of 15 years (2000 to 2014). Figure 9 shows flow-duration curves for select burned sub-basins as well as for the entire study watershed and Table 5 lists the ecosurplus and ecodeficit values associated with each computed flow-duration curve. Flow-duration curves were generated using total runoff, which includes both surface and subsurface water fluxes leaving the sub-basin or watershed. The ecosurplus and ecodeficit metrics are a dimensionless measure which represent the overall loss (ecodeficit) and gain (ecosurplus) in streamflow (Vogel et al., 2007) between scenarios.

Similar to findings from the hydrologic budgets, it is evident that flow-duration curves change under wildfire conditions on the sub-basin scale. Also, little change is seen at the watershed scale (Figure 9 and Table 5). This is perhaps the result of wildfire effects at the watershed scale being damped by non-burned portions of the contributing area.

Figure 9 also suggests that wildfire has little impact on flow-duration curves for areas with low total burn area percentages, but seems to impact flow-duration curves for area with higher total burn area percentages. For example, in sub-basins 30 we see that less frequent streamflows become greater in magnitude under the wildfire scenario (i.e. we see an ecosurplus). Whereas, in sub-basin 19 (a less burned area) we see little change in the flow-duration curve. Previous research efforts have involved a paired-catchment analysis to compare flow duration curves for pre and post-wildfire conditions (Liu et al., 2004; Newtson, 2013). Both Newtson (2013) and Liu et al. (2004) found a general increase in percentile streamflow as a result of wildfire. However, Liu et al. (2004) examined precipitation duration curves for the study areas and concluded that changes in precipitation between locations explained the difference in streamflow and not necessarily wildfire. For this study, the two scenarios approach uses an identical precipitation record for both scenarios. Thus, the study eliminates limitations associated with temporal and special variation in precipitation. Table 5 indicates the streamflows for the burned sub-basins appear to be ecosurplus versus ecodeficit when the wildfire scenario is compared with the no-wildfire scenario. The ecosurplus values range from 0.004 to 0.279. Kannan and Jeong (2011) indicate that for high streamflows a large ecosurplus is likely to have moderate to high impacts to stream health. In this case, the ecosurplus values associated with the heavily burned sub-basins (i.e., sub-basins 28, 30, 26, and 32) are much greater in magnitude when compared to the other ecosuplus values. Thus, impacts to stream health are expected to be the greatest in heavily burned areas.




### 3.5 Limitations and future work

Figure 10 displays simulated versus observed monthly streamflows as well as average monthly simulated and observed streamflow for the Mouth of Canyon. This figure suggests that the model slightly overestimates larger monthly streamflows: specifically, those during the month of June when streamflows are elevated due to mountain snowpack melting. Also, the model appears to slightly underestimate streamflows during late summer into autumn. These systematic errors may be due to SWAT releasing snowmelt too quickly during spring runoff, thus, rising streamflows are simulated earlier than observations during the melting season. Further, perhaps the tendency of the model to simulate earlier snowmelt results in higher simulated streamflow during the latter part of summer and early autumn. This deficiency may be the result of SWAT misrepresenting snowmelt processes or perhaps faulty model parameterization. Thus, it is thought that hydrologic model uncertainty is introduced here and it is recommended that additional research be focused on better representing snowmelt processes in mountainous watersheds.

### 4 Conclusions

Long term simulation scenario analysis at the sub-basin and watershed scales was used to characterize hydrologic response to wildfires in mountainous regions. This was achieved by applying the hydrologic model SWAT to a watershed recently exposed to significant wildfire incident located in northern Colorado, USA. The model represents pre-wildfire and post-wildfire conditions by implementing the SWAT land use change module during simulations to represent burned area as a result of wildfire. Geospatial data representing LULC, soil, terrain, and climate attributes of the study watershed was used to develop the model. An optimal parameter set was obtained for pre-wildfire and post-wildfire conditions through the automated DDS optimization algorithm. Error statistics were calculated to evaluate model performance with regard to daily observed naturalized streamflows. Results indicate a good model performance, with an $E_{NS}$ of 0.82 during calibration as well as 0.71 and 0.88 for the no-wildfire and wildfire testing periods, respectively, for daily streamflows at the Mouth of Canyon. No-wildfire and wildfire scenarios representing a 15 year (2000 to 2014) simulation period were created from the optimal parameter set achieved during model calibration. These scenarios were used to characterize the hydrologic response to wildfires.

Specific objectives of this study were to investigate changes in average annual total runoff volume, average annual hydrologic budgets, and flow-duration curves across multiple scales as a result of wildfire. At the watershed scale, wildfire conditions appear to have little effect on the hydrologic responses with the exception of total runoff volume. However, at the sub-basin scale, simulations suggest that wildfire effects trend with burned area upstream. A total runoff increase up to approximately 75 percent between scenarios was found. Generally, water budgets showed more surface runoff versus subsurface flow, which suggests infiltration rates decrease under post-wildfire conditions. Flow-duration curves for burned sub-basins showed that less frequent streamflows become greater in magnitude leading to ecosurplus values of up to 0.279.



Results reported in this study show an overall acceptable performance of the SWAT model in simulating daily streamflows under pre and post-wildfire conditions to characterize the hydrologic response to wildfires. However, this method required comprehensive knowledge of the watershed, was time consuming, and was computationally intensive. Further, this study demonstrates the need for improvement in understanding the rainfall-runoff prediction relationship for burned areas.

**Acknowledgements**

This project was funded by the United States Department of Agriculture National Institute of Food and Agriculture (USDA-NIFA) grant number: 2012-67003-19904.

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



Table 1. SWAT model input data.

| Data type | Data used | Description |
|---|---|---|
| Terrain | Digital Elevation Model | National Elevation Dataset \| 1/3 arc-second (~10 m) |
| Land Use / Land Cover | 2011 Land Cover | National Land Cover Dataset \| 30 m |
| Burn Severity | Thematic Burn Severity Delineation | Monitoring Trends in Burn Severity High Park Fire Assessment \| 30 m |
| Soil | Soil Map Unit Delineation | Gridded Soil Survey Geographic Database for Colorado and Wyoming \| 10 m |
| Meteorological | Precipitation and Temperature Measurements | Global Historical Climatology Network Database \| Daily |
| Streamflow | Naturalized Streamflow Data | Northern Colorado Water Conservancy District \| Daily |
| Model Parameters | SWAT Model Databases | Land Cover Land Use, Soil, and Weather Parameters |

Table 2. Error statistics between observed and simulated daily streamflows for the calibration period as well as the testing periods. Performance ratings based on Motovilov (1999).

| Simulation | Simulation period | Relative error | Nash-Sutcliffe efficiency | Performance rating |
|---|---|---|---|---|
| Pre-wildfire testing | 2000-2004 | -19.52 | 0.71 | Satisfactory |
| Calibration | 2005-2013 | 1.68 | 0.82 | Good |
| Post-wildfire testing | 2014 | 9.31 | 0.88 | Good |
| All | 2000-2014 | -2.73 | 0.82 | Good |

Table 3. Error statistics between observed and simulated monthly streamflows for the calibration period as well as the testing periods. Performance ratings based on Moriasi and Arnold (2007).

| Simulation | Simulation period | Relative error | Nash-Sutcliffe efficiency | Performance rating |
|---|---|---|---|---|
| Pre-wildfire testing | 2000-2004 | -19.36 | 0.80 | Very Good |
| Calibration | 2005-2013 | 1.77 | 0.88 | Very Good |
| Post-wildfire testing | 2014 | 9.42 | 0.96 | Very Good |
| All | 2000-2014 | -2.61 | 0.89 | Very Good |



Table 4. Average annual total runoff volumes and depths for both the no-wildfire and fire scenarios, shown for the burned sub-basins as well as for the entire study watershed. Area is also include for reference.

| Sub-basin | Area (km$^2$) | Average annual total runoff volume (mega m$^3$/yr) | | Average annual total runoff depth (mm/yr) | |
|---|---|---|---|---|---|
| | | No-wildfire | Wildfire | No-wildfire | Wildfire |
| 19 | 89.56 | 1.82 | 2.10 | 20.4 | 23.4 |
| 24 | 56.53 | 0.74 | 1.01 | 13.1 | 17.9 |
| 25 | 5.41 | 0.14 | 0.14 | 25.4 | 25.7 |
| 26 | 17.39 | 0.61 | 0.98 | 35.0 | 56.4 |
| 28 | 14.64 | 0.33 | 0.58 | 22.8 | 39.8 |
| 29 | 47.15 | 1.59 | 1.67 | 33.7 | 35.3 |
| 30 | 106.95 | 4.16 | 6.81 | 38.9 | 63.7 |
| 32 | 10.86 | 0.30 | 0.49 | 27.4 | 45.4 |
| 35 | 269.11 | 38.91 | 41.70 | 144.6 | 154.9 |
| Study Watershed | 2,732 | 323.52 | 330.38 | 118.5 | 121.1 |

Table 5. Ecosurplus and ecodeficit values for the burned sub-basins as well as for the entire study watershed.

| Sub-basin | Ecosurplus | Ecodeficit |
|---|---|---|
| 19 | 0.065 | 0.001 |
| 24 | 0.100 | 0.004 |
| 25 | 0.004 | 0.000 |
| 26 | 0.168 | 0.011 |
| 28 | 0.248 | 0.010 |
| 29 | 0.089 | 0.000 |
| 30 | 0.279 | 0.016 |
| 32 | 0.157 | 0.010 |
| 35 | 0.093 | 0.001 |
| Study Watershed | 0.093 | 0.001 |

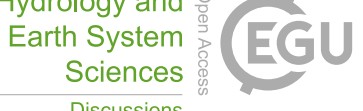



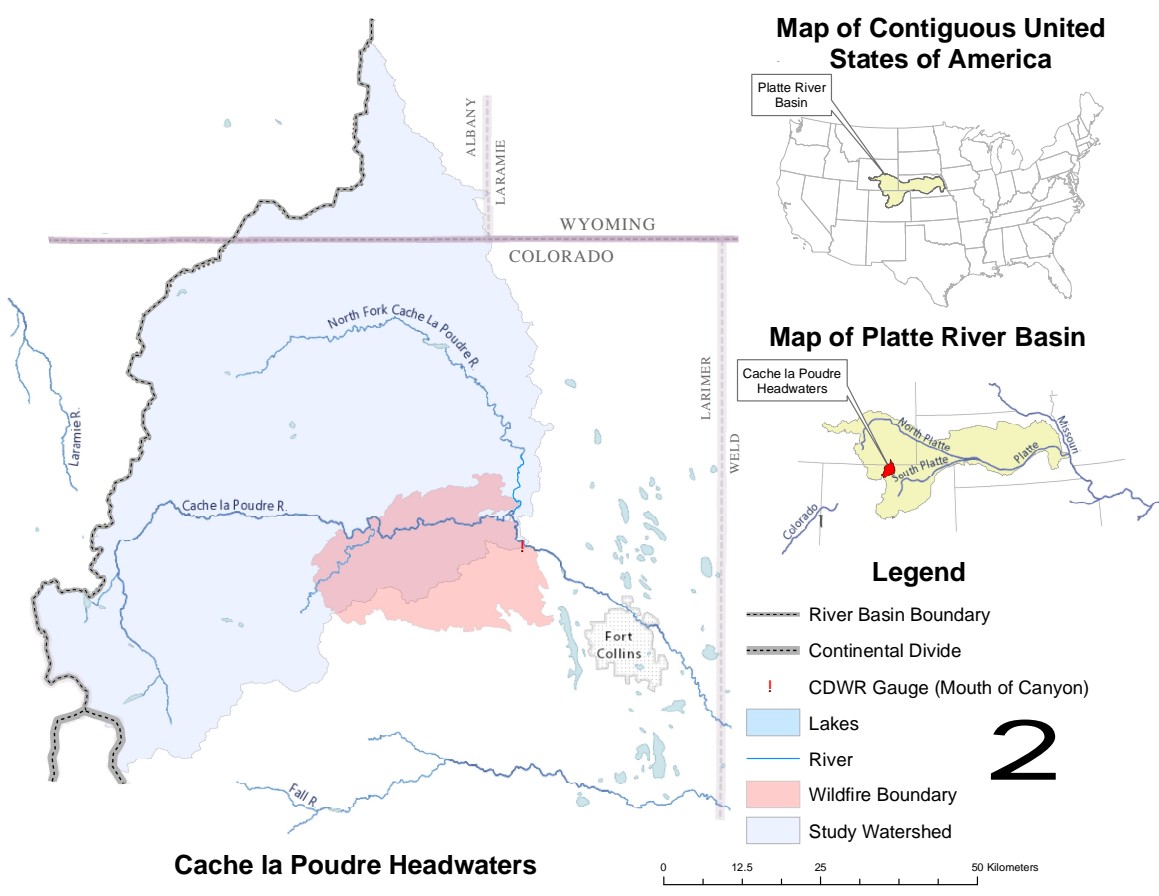

Figure 1: Study area map which includes the location of study watershed and the CDWR surface water gauge.



Figure 2: Initial SWAT model development summary. Outer figures show terrain, LULC, wildfire burn severity, soils, and HRUs. Main figure (lower right) includes labeled sub-basins, location of meteorological stations, and reach network. Note that for illustrative purposes the soils and LULC classifications shown are simplified versions of the actual classifications used to establish HRUs.





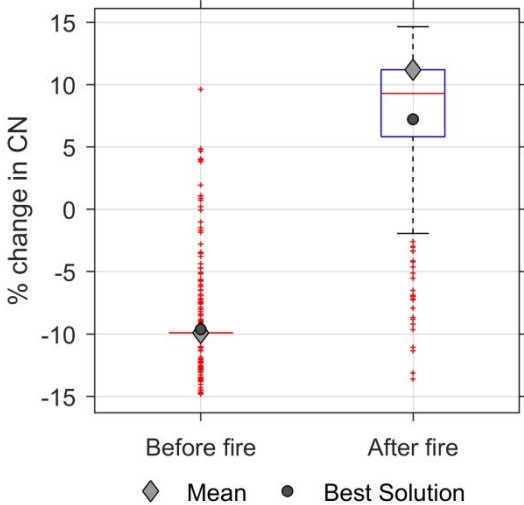

Figure 3: Boxplots showing the range of % change in CNs for before and after fire conditions (the boxes show the range of values between 25th and 75th percentile; the whiskers show the 0.5 and 99.5 percentile).

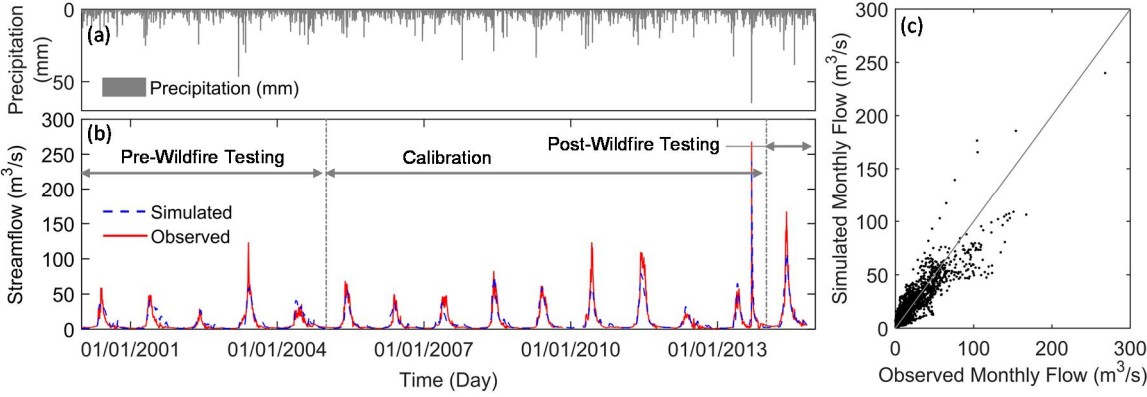

Figure 4: (a) Total daily precipitation during simulation period. (b) Observed versus simulated average daily streamflow hydrographs. (c) Observed versus simulated average daily streamflows scatter plot.



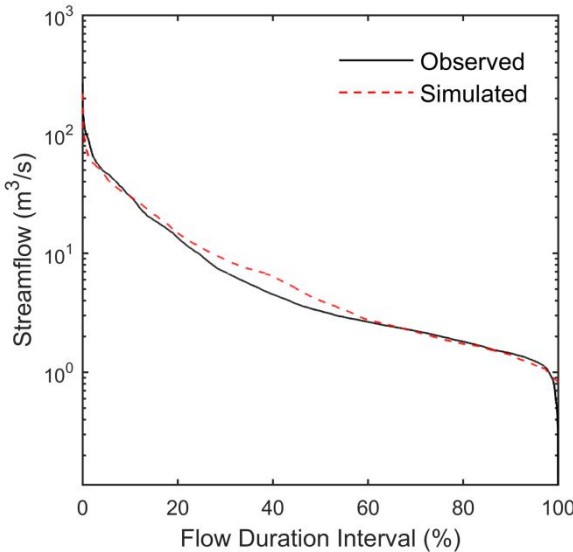

Figure 5: Flow-duration curve at the Mouth of Canyon for the entire simulation period.

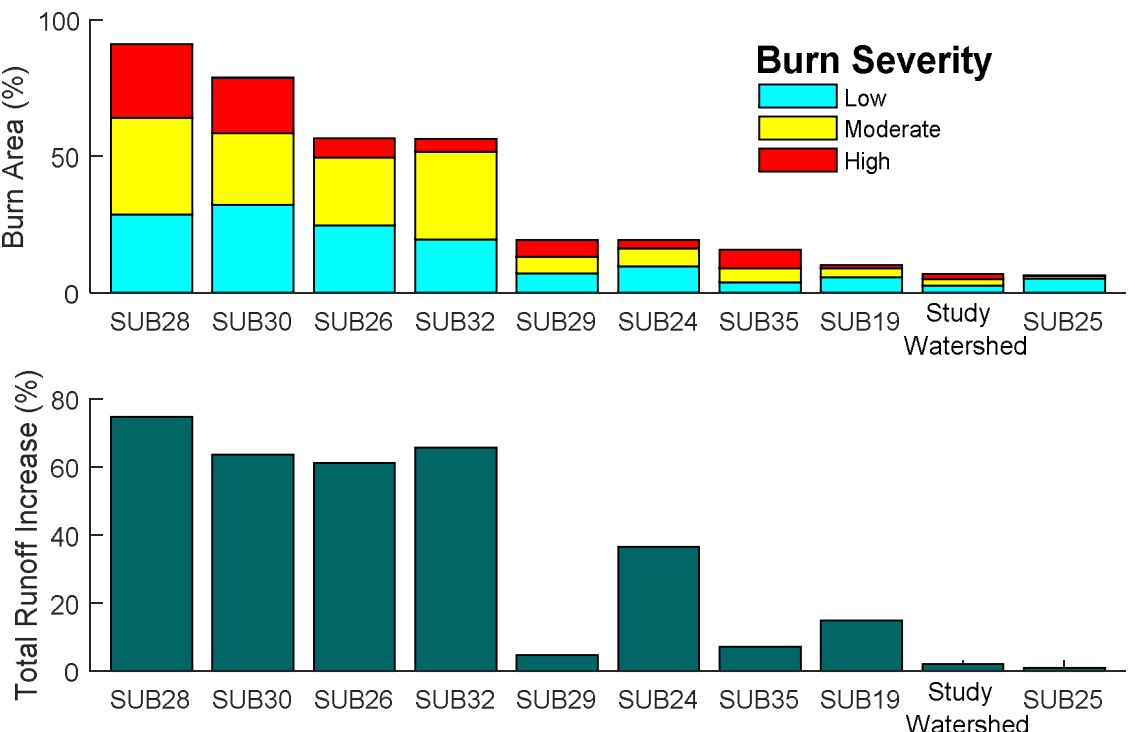




Figure 6: Burn severity distribution (above) and average annual total runoff percent increase between the no-wildfire and fire scenarios (below). Results are shown for the burned sub-basins as well as for the entire study watershed ("Study Watershed') arranged in descending order from left to right based on total percent burned area.

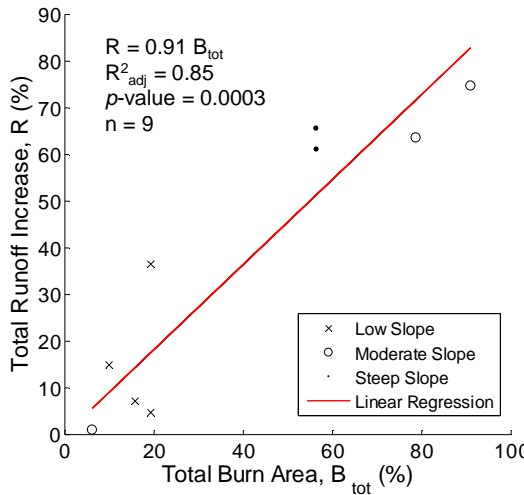

5    Figure 7: Linear regression model fitted between the total runoff volume increase and total burn area percentage. Catchment slope is categorized as low (slope < 0.30), moderate (0.30 ≤ slope < 0.40), and steep (slope ≥ 0.40) for each sub-basin.





Figure 8: Hydrologic budgets showing the fate of average annual precipitation (i.e., evapotranspiration, total runoff, and other) with the fate of average annual total runoff (i.e., surface and subsurface) for select sub-basins and the entire study watershed.





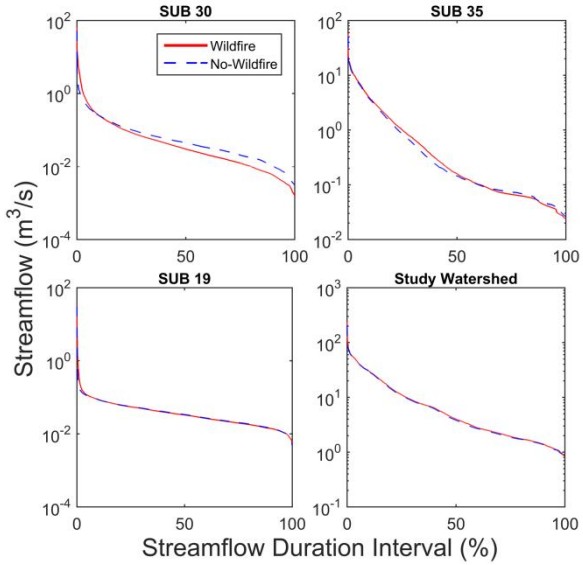

Figure 9: Flow-duration curves for select sub-basin as well as the entire study watershed. Subbasin area and percentage of burned area for subbasins 30; 35; 19 and study watersheds are: 11 km2, 79%; 269 km2, 16%; 90 km2, 10% and 2,732 km2, 14% respectively.

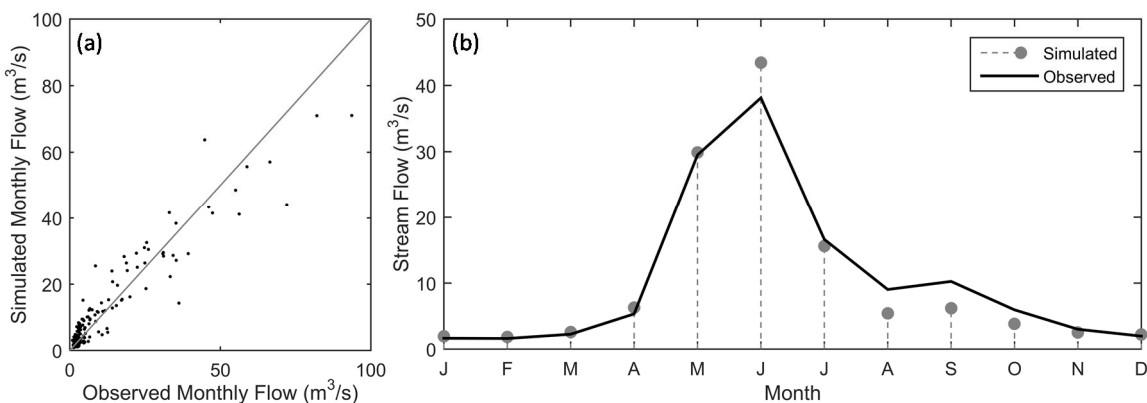

Figure 10: (a) Scatter plot of simulated versus observed monthly streamflows and (b) the observed versus simulated average monthly streamflows for the simulation period.



## Appendix A.

### A.1. Detailed Description of Model Data

The 10 m resolution Digital Elevation Model (DEM), courtesy of the United States Geological Survey (USGS) National Elevation Dataset (USGS TNM, 2016), was used to describe the topography within the watershed. The study watershed
ranges in elevation from 4,138 m at the Continental Divide down to 1,493 m at the Mouth of Canyon. The distribution of elevation within the study watershed is displayed in Fig. A1.

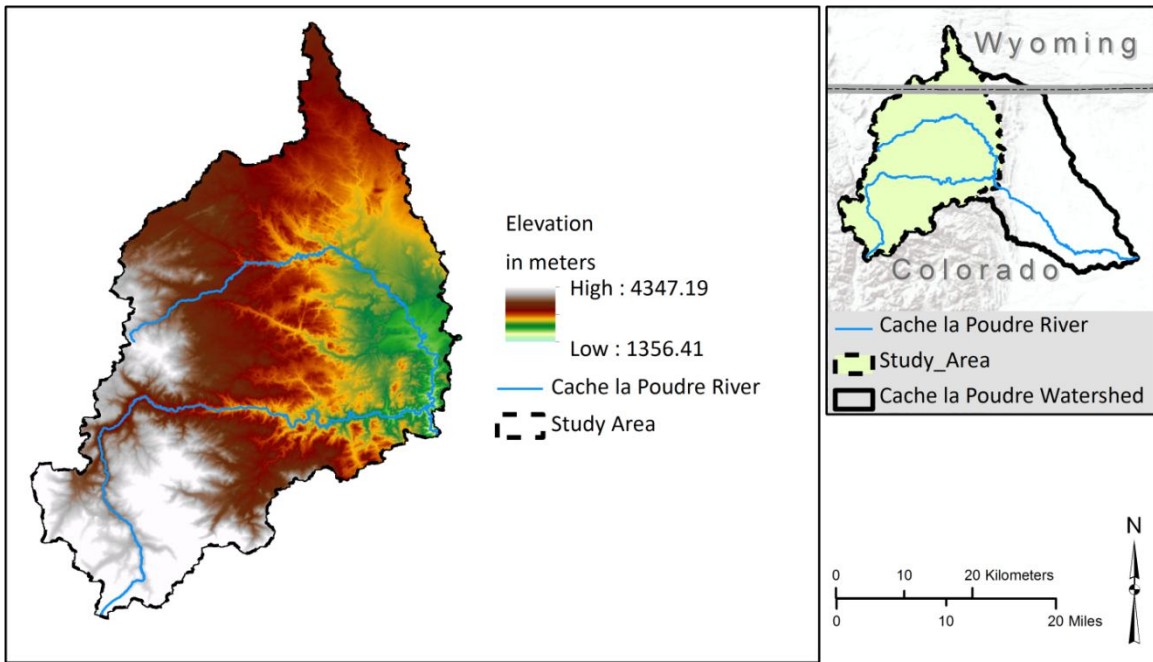

Fig. A1. Distribution of elevation within the study watershed based on the 10 m DEM.

The 30 m resolution National Land Cover Database (NLCD) 2011 Land Cover dataset created through a project conducted
by the Multi-Resolution Land Characteristics (MRLC) Consortium was used to describe the LULC distribution for the study watershed (USGS TNM, 2016). NLCD 2011 Land Cover uses 16 classifications that are based primarily on an analysis of circa 2011 Landsat imagery. Distribution of the major types found within the study watershed may be seen in Fig. A2 and a complete breakdown is shown in Table A1. Generally, the study watershed consists of forest (primary evergreen type) with considerably large portions covered by shrubland and herbaceous vegetation. Note the study watershed is relatively
undeveloped, with less than 1 percent of the land surface developed for commercial, industrial, or residential purposes. Through comparison of earlier NLCD products, it is evident that LULC changes little between the years 2000, 2006, and 2011. Therefore, it was assumed appropriate to use NLCD 2011 Land Cover for the entire simulation period. A



comprehensive LULC change analysis for the study watershed using NLCD 2000, 2006, and 2011 Land Cover is included in Table A1.

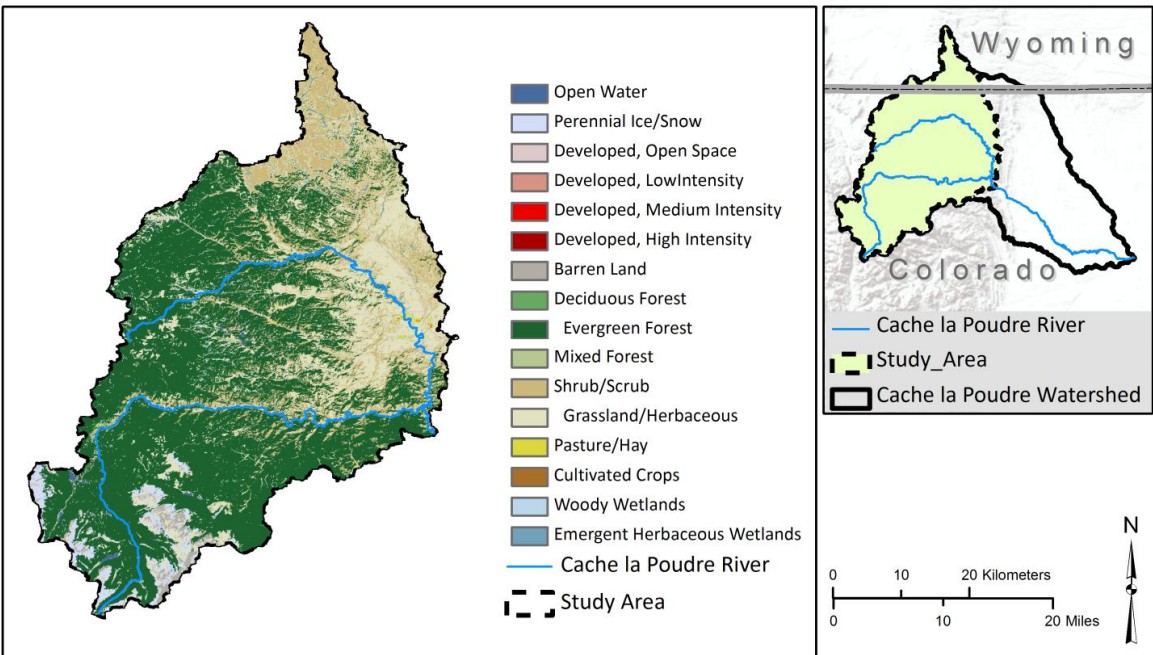

Fig. A2. Distribution of major LULC types in study watershed based on MRLC's NLCD 2011 Land Cover dataset.

5          Table A1. Comprehensive distribution of LULC in study watershed based on NLCD 2001, 2006, and 2011.

| Class | Description | Portion of study watershed (%) | | |
|---|---|---|---|---|
| | | 2001 | 2006 | 2011 |
| Water | Open Water | 0.30 | 0.28 | 0.29 |
| Water | Perennial Ice/Snow | 2.27 | 2.27 | 2.27 |
| Developed | Developed, Open Space | 0.57 | 0.57 | 0.57 |
| Developed | Developed, Low Intensity | 0.17 | 0.17 | 0.17 |
| Developed | Developed, Medium Intensity | 0.01 | 0.01 | 0.01 |
| Developed | Developed, High Intensity | 0.00 | 0.00 | 0.00 |
| Barren | Barren Land (Rock/Sand/Clay) | 1.36 | 1.36 | 1.36 |
| Forest | Deciduous Forest | 0.58 | 0.58 | 0.57 |
| Forest | Evergreen Forest | 56.17 | 56.07 | 56.00 |
| Forest | Mixed Forest | 0.04 | 0.04 | 0.04 |
| Shrubland | Shrub/Scrub | 17.59 | 17.69 | 17.76 |
| Herbaceous | Grassland/Herbaceous | 18.76 | 18.79 | 18.79 |



| Planted/Cultivated | Pasture/Hay | 0.24 | 0.24 | 0.24 |
| Planted/Cultivated | Cultivated Crops | 0.01 | 0.01 | 0.01 |
| Wetlands | Woody Wetlands | 1.49 | 1.50 | 1.50 |
| Wetlands | Emergent Herbaceous Wetlands | 0.44 | 0.43 | 0.43 |

Burned areas within the watershed were identified using the High Park Wildfire Assessment (Monitoring Trends in Burn Severity Project, 2014) conducted as a part of the Monitoring Trends in Burn Severity (MTBS) project directed by groups within the USGS and United States Forest Service. The MTBS project was introduced to consistently map burn severity and

5   boundaries of wildfires across all lands of the USA from 1984 and beyond. The product of this assessment includes a Thematic Burn Severity Delineation which depicts severity as unburned to low, low, moderate, high, and increased greenness (i.e., increase post-wildfire vegetation response). Through examining the wildfire boundary, it is evident that the High Park Wildfire Assessment includes the Hewlett wildfire which occurred just prior to the High Park wildfire. The burn severity distribution of the Hewlett and High Park wildfire within the study watershed may be seen in Fig. A3. The

10   distribution of the different burn severities within the wildfire boundary is relatively even.

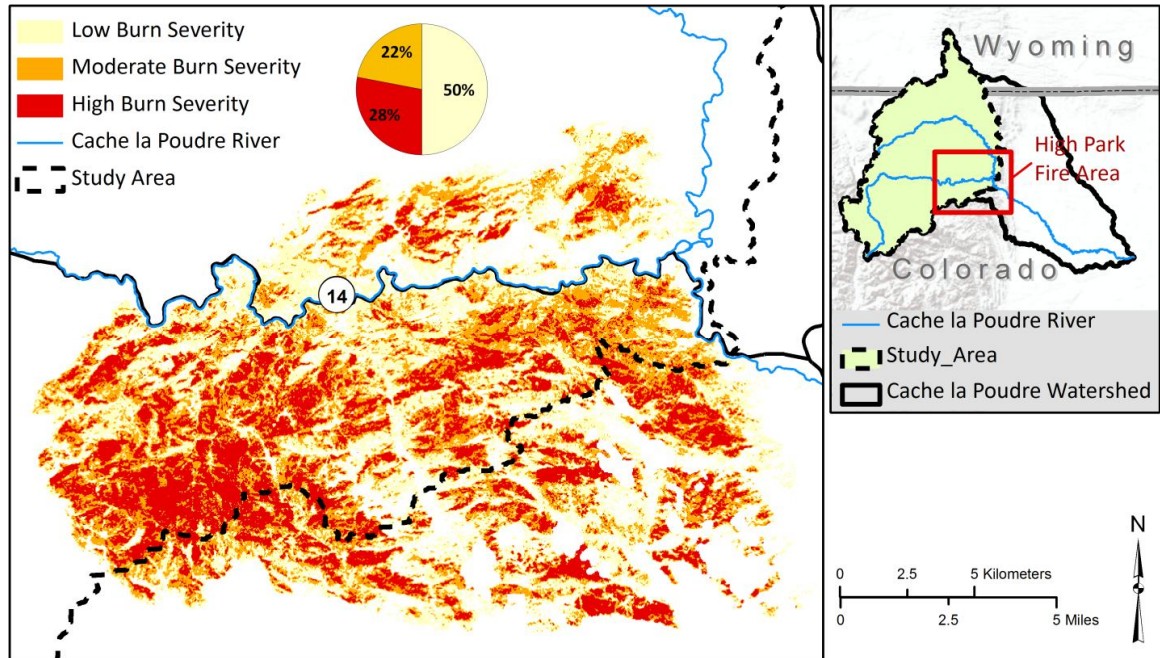

Fig. A3. Distribution of burn severity of the Hewlett and High Park wildfires within the study watershed based on MTSB's High Park Fire Assessment.





The Gridded Soil Survey Geographic (gSSURGO) database for Colorado and Wyoming (Soil Survey Staff, 2015), obtained from the USDA Natural Resources Conservation Service (NRCS), was used to represent the distribution of soil within the study watershed. This dataset contains soil mapping, which includes outlined areas called map units. These map units have unique properties, interpretations, and productivity which describe the soils. The study watershed contains 153 different map

units. The SWAT SSURGO Soils database (U.S. Department of Agriculture Agricultural Research Service, 2012) was used to describe various model parameters for each gSSURGO map unit. One model parameter of particular interest is the Hydrologic Soil Group (HSG). The HSG is a classification established by the NRCS which is based on the runoff potential of a given soil. This classification consists of four groups: A, B, C, and D. Generally, soils designated as type A have the smallest runoff potential and soils designated as type D have the greatest. The distribution of soil as represented by HSG

within the study watershed is shown in Fig. A4. Generally, the study watershed consists of Hydrologic Soil Group D type soils, indicating the area has very low to moderate infiltration rates. This implies that the study watershed may have a high runoff potential.

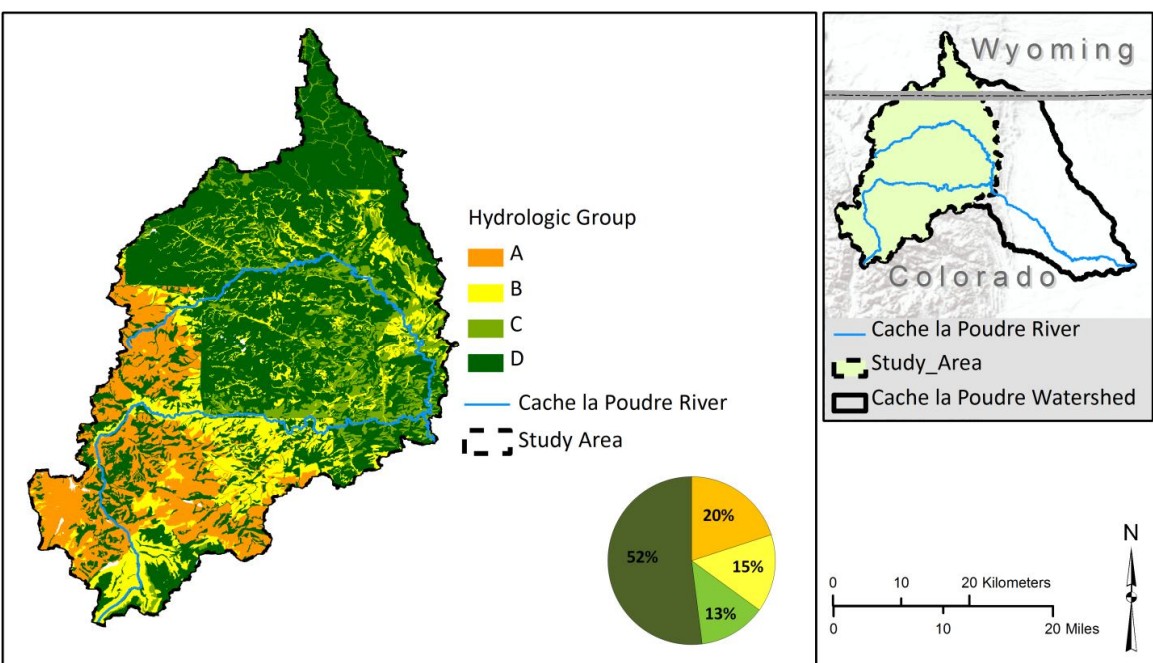

Fig. A4. Distribution of soil as represented by Hydrologic Soil Groups A-D within the study watershed based on the
USDA's gSSURGO database.

Daily measurements of precipitation, maximum temperature, and minimum temperature for the study watershed were obtained from the Global Historical Climatology Network (GHCN) Daily dataset (NOAA, 2016), which is maintained by the National Climatic Data Center (NCDC). The NCDC extensively quality assures GHCN daily data prior to data release. This



is accomplished using a multi-tiered approach including a formatting check as well as a quality test looking for a variety of data problems. Based on this, no further quality control beside removal of flagged data was conducted. The stations were selected based on location, type of data provided, length of record, and completeness of record. A complete list of stations may be found in Table A2. Mean annual precipitation ranges from 330 mm at the lower elevations to 1350 mm at the higher

elevations and mean annual temperature ranges from approximately 9° C at the lower elevations to -5° C at the higher elevations.

Table A2. Meteorological stations used for this study.

| Station name | Latitude | Longitude | Elevation (m) | Notes |
|---|---|---|---|---|
| STOVE PRAIRIE 2 WNW CO US | 40.6263 | -105.391 | 2357.9 | Precip. only |
| RED FEATHER 5.9 NE CO US | 40.86 | -105.509 | 2414.9 | Precip. only |
| BLV 4.0 NW CO US | 40.6754 | -105.215 | 1631.9 | Precip. only |
| BUCKHORN MOUNTAIN 1 E CO US | 40.6167 | -105.283 | 2255.5 | |
| HOURGLASS RESERVOIR CO US | 40.5831 | -105.632 | 2901.7 | |
| RUSTIC 9 WSW CO US | 40.7167 | -105.717 | 2347 | |
| VIRGINIA DALE 7 ENE CO US | 40.9656 | -105.219 | 2138.2 | |
| RED FEATHER COLORADO CO US | 40.7981 | -105.572 | 2499.4 | Temp. only |
| DEADMAN HILL CO US | 40.8 | -105.767 | 3115.1 | |
| JOE WRIGHT CO US | 40.5333 | -105.883 | 3084.6 | |
| WILLOW PARK CO US | 40.4333 | -105.733 | 3261.4 | |

Precipitation within the study watershed is greatest during the winter months. Snow accumulates which generates the

mountain snowpack that is then released during the spring and early summer months. In an effort to support economic, environmental, and recreational water demands downstream, manmade structures such as diversions, storage reservoirs, and irrigation canals are used to store and distribute the snowmelt runoff during times of the year when the demand of water exceeds its availability. Thus, the Poudre River flow regime is modified. One study of the Poudre watershed described several flow regime modifications including delayed hydrograph rise, decreased peak streamflows, and lower winter base

flows (Richer, 2009). In an effort to ensure hydrologic processes are represented appropriately, naturalized streamflows were used for model calibration and testing. Naturalized streamflows remove the influence of afore mentioned features such as diversions and impoundments. Daily naturalized streamflows were collected from Northern Colorado Water Conservancy District at the Mouth of Canyon. Fig. A5 shows the relationship between naturalized daily average streamflow versus observed daily average streamflow.





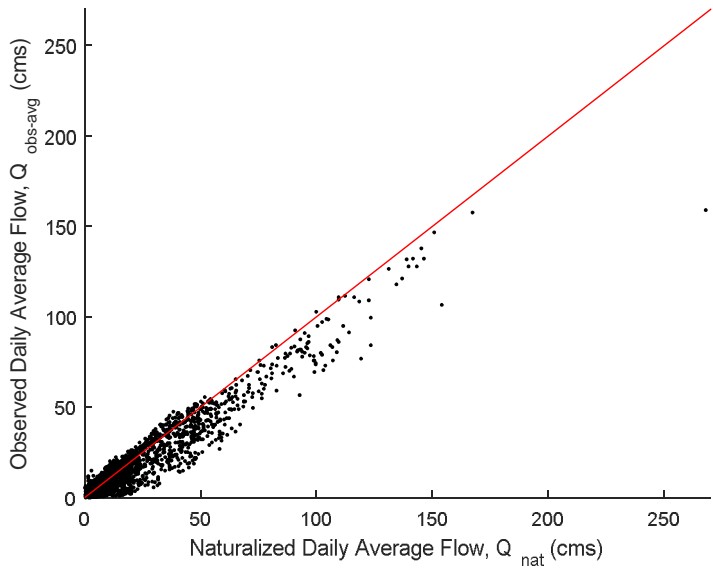

Fig. A5. CDWR naturalized daily average streamflow versus observed daily average streamflow with 1 to 1 reference line.



## A.2. Supplementary Tables

Table A3. Original SWAT database land use / land cover lookup table.

| NLCD code | NLCD description | SWAT code | SWAT LULC description |
|---|---|---|---|
| 11 | Open Water | WATR | Water |
| 12 | Perennial Ice/Snow | WATR | Water |
| 21 | Developed, Open Space | URLD | Residential-Low Density |
| 22 | Developed, Low Intensity | URMD | Residential-Medium Density |
| 23 | Developed, Medium Intensity | URHD | Residential-High Density |
| 24 | Developed, High Intensity | UIDU | Industrial |
| 31 | Barren Land (Rock/Sand/Clay) | SWRN | Southwestern US (Arid) Range |
| 32 | Unconsolidated Shore | SWRN | Southwestern US (Arid) Range |
| 41 | Deciduous Forest | FRSD | Forest-Deciduous |
| 42 | Evergreen Forest | FRSE | Forest-Evergreen |
| 43 | Mixed Forest | FRST | Forest-Mixed |
| 51 | Dwarf Scrub | RNGB | Range-Brush |
| 52 | Shrub/Scrub | RNGB | Range-Brush |
| 71 | Grassland/Herbaceous | RNGE | Range-Grasses |
| 72 | Sedge/Herbaceous | RNGE | Range-Grasses |
| 73 | Lichens | RNGE | Range-Grasses |
| 74 | Moss | RNGE | Range-Grasses |
| 81 | Pasture/Hay | HAY | Hay |
| 82 | Cultivated Crops | AGRR | Agricultural Land-Row Crops |
| 90 | Woody Wetlands | WETF | Wetlands-Forested |
| 91 | Palustrine Forested Wetland | WETF | Wetlands-Forested |
| 92 | Palustrine Scrub/Shrub Wetland | WETL | Wetlands-Mixed |
| 93 | Estuarine Forested Wetland | WETF | Wetlands-Forested |
| 94 | Estuarine Scrub/Shrub Wetland | WETL | Wetlands-Mixed |
| 95 | Emergent Herbaceous Wetlands | WETN | Wetlands-Non-Forested |
| 96 | Palustrine Emergent Wetland (Persistent) | WETN | Wetlands-Non-Forested |
| 97 | Estuarine Emergent Wetland* | WETN | Wetlands-Non-Forested |
| 98 | Palustrine Aquatic Bed | WATR | Water |
| 99 | Estuarine Aquatic Bed | WATR | Water |

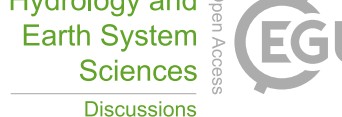
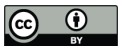

Table A4. Pre-wildfire editied lookup table and corresponding curve numbers.

| Code | NLCD description | SWAT code | SWAT LULC description | CN2A | CN2B | CN2C | CN2D |
|---|---|---|---|---|---|---|---|
| 111 | Open Water | WATR | Water | - | - | - | - |
| 121 | Developed, Open Space | PFLA | Pre-Fire Residential-Low Density Low Burn | 31 | 59 | 72 | 79 |
| 122 | Developed, Low Intensity | PFLB | Pre-Fire Residential-Medium Density Low Burn | 31 | 59 | 72 | 79 |
| 123 | Developed, Medium Intensity | PFLC | Pre-Fire Residential-High Density Low Burn | 31 | 59 | 72 | 79 |
| 141 | Deciduous Forest | PFLD | Pre-Fire Forest-Deciduous Low Burn | 45 | 66 | 77 | 83 |
| 142 | Evergreen Forest | PFLE | Pre-Fire Forest-Evergreen Low Burn | 25 | 55 | 70 | 77 |
| 143 | Mixed Forest | PFLF | Pre-Fire Forest-Mixed Low Burn | 36 | 60 | 73 | 79 |
| 152 | Shrub/Scrub | PFLG | Pre-Fire Range-Brush Low Burn | 39 | 61 | 74 | 80 |
| 171 | Grassland/Herbaceous | PFLH | Range-Grasses Low Burn | 49 | 69 | 79 | 84 |
| 181 | Pasture/Hay | PFLI | Pre-Fire Hay Low Burn | 31 | 59 | 72 | 79 |
| 190 | Woody Wetlands | PFLJ | Pre-Fire Wetlands-Forested Low Burn | 45 | 66 | 77 | 83 |
| 195 | Emergent Herbaceous Wetlands | PFLK | Pre-Fire Wetlands-Non-Forested Low Burn | 49 | 69 | 79 | 84 |
| 211 | Open Water | WATR | Water | - | - | - | - |
| 221 | Developed, Open Space | PFML | Pre-Fire Residential-Low Density Moderate Burn | 31 | 59 | 72 | 79 |
| 241 | Deciduous Forest | PFMM | Pre-Fire Forest-Deciduous Moderate Burn | 45 | 66 | 77 | 83 |
| 242 | Evergreen Forest | PFMN | Pre-Fire Forest-Evergreen Moderate Burn | 25 | 55 | 70 | 77 |
| 243 | Mixed Forest | PFMO | Pre-Fire Forest-Mixed Moderate Burn | 36 | 60 | 73 | 79 |
| 252 | Shrub/Scrub | PFMP | Pre-Fire Range-Brush Moderate Burn | 39 | 61 | 74 | 80 |
| 271 | Grassland/Herbaceous | PFMQ | Pre-Fire Range-Grasses Moderate Burn | 49 | 69 | 79 | 84 |
| 290 | Woody Wetlands | PFMR | Pre-Fire Wetlands-Forested Moderate Burn | 45 | 66 | 77 | 83 |
| 295 | Emergent Herbaceous Wetlands | PFMS | Pre-Fire Wetlands-Non-Forested Moderate Burn | 49 | 69 | 79 | 84 |
| 321 | Developed, Open Space | PFHT | Pre-Fire Residential-Low Density High Burn | 31 | 59 | 72 | 79 |
| 341 | Deciduous Forest | PFHU | Pre-Fire Forest-Deciduous High Burn | 45 | 66 | 77 | 83 |
| 342 | Evergreen Forest | PFHV | Pre-Fire Forest-Evergreen High Burn | 25 | 55 | 70 | 77 |
| 343 | Mixed Forest | PFHW | Pre-Fire Forest-Mixed High Burn | 36 | 60 | 73 | 79 |
| 352 | Shrub/Scrub | PFHX | Pre-Fire Range-Brush High Burn | 39 | 61 | 74 | 80 |
| 371 | Grassland/Herbaceous | PFHY | Pre-Fire Range-Grasses High Burn | 49 | 69 | 79 | 84 |
| 390 | Woody Wetlands | PFHZ | Pre-Fire Wetlands-Forested High Burn | 45 | 66 | 77 | 83 |





Table A5. Post-wildfire edited lookup table and corresponding curve numbers.

| Code | NLCD description | SWAT code | SWAT LULC description | CN2A | CN2B | CN2C | CN2D |
|---|---|---|---|---|---|---|---|
| 111 | Open Water | WATR | Water | - | - | - | - |
| 121 | Developed, Open Space | FRLA | Post-Fire Residential-Low Density Low Burn | 36 | 64 | 77 | 84 |
| 122 | Developed, Low Intensity | FRLB | Post-Fire Residential-Medium Density Low Burn | 36 | 64 | 77 | 84 |
| 123 | Developed, Medium Intensity | FRLC | Post-Fire Residential-High Density Low Burn | 36 | 64 | 77 | 84 |
| 141 | Deciduous Forest | FRLD | Post-Fire Forest-Deciduous Low Burn | 50 | 71 | 82 | 88 |
| 142 | Evergreen Forest | FRLE | Post-Fire Forest-Evergreen Low Burn | 30 | 60 | 75 | 82 |
| 143 | Mixed Forest | FRLF | Post-Fire Forest-Mixed Low Burn | 41 | 65 | 78 | 84 |
| 152 | Shrub/Scrub | FRLG | Post-Fire Range-Brush Low Burn | 44 | 66 | 79 | 85 |
| 171 | Grassland/Herbaceous | FRLH | Post-Fire Range-Grasses Low Burn | 54 | 74 | 84 | 89 |
| 181 | Pasture/Hay | FRLI | Post-Fire Hay Low Burn | 36 | 64 | 77 | 84 |
| 190 | Woody Wetlands | FRLJ | Post-Fire Wetlands-Forested Low Burn | 50 | 71 | 82 | 88 |
| 195 | Emergent Herbaceous Wetlands | FRLK | Post-Fire Wetlands-Non-Forested Low Burn | 54 | 74 | 84 | 89 |
| 211 | Open Water | WATR | Water | - | - | - | - |
| 221 | Developed, Open Space | FRML | Post-Fire Residential-Low Density Moderate Burn | 41 | 69 | 82 | 89 |
| 241 | Deciduous Forest | FRMM | Post-Fire Forest-Deciduous Moderate Burn | 55 | 76 | 87 | 93 |
| 242 | Evergreen Forest | FRMN | Post-Fire Forest-Evergreen Moderate Burn | 35 | 65 | 80 | 87 |
| 243 | Mixed Forest | FRMO | Post-Fire Forest-Mixed Moderate Burn | 46 | 70 | 83 | 89 |
| 252 | Shrub/Scrub | FRMP | Post-Fire Range-Brush Moderate Burn | 49 | 71 | 84 | 90 |
| 271 | Grassland/Herbaceous | FRMQ | Post-Fire Range-Grasses Moderate Burn | 59 | 79 | 89 | 94 |
| 290 | Woody Wetlands | FRMR | Post-Fire Wetlands-Forested Moderate Burn | 55 | 76 | 87 | 93 |
| 295 | Emergent Herbaceous Wetlands | FRMS | Post-Fire Wetlands-Non-Forested Moderate Burn | 59 | 79 | 89 | 94 |
| 321 | Developed, Open Space | FRHT | Post-Fire Residential-Low Density High Burn | 46 | 74 | 87 | 94 |
| 341 | Deciduous Forest | FRHU | Post-Fire Forest-Deciduous High Burn | 60 | 81 | 92 | 98 |
| 342 | Evergreen Forest | FRHV | Post-Fire Forest-Evergreen High Burn | 40 | 70 | 85 | 92 |
| 343 | Mixed Forest | FRHW | Post-Fire Forest-Mixed High Burn | 51 | 75 | 88 | 94 |
| 352 | Shrub/Scrub | FRHX | Post-Fire Range-Brush High Burn | 54 | 76 | 89 | 95 |
| 371 | Grassland/Herbaceous | FRHY | Post-Fire Range-Grasses High Burn | 64 | 84 | 94 | 99 |
| 390 | Woody Wetlands | FRHZ | Post-Fire Wetlands-Forested High Burn | 60 | 81 | 92 | 98 |



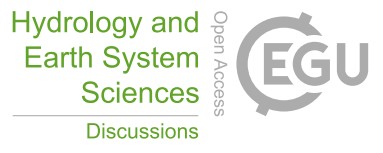

Table A6. SWAT calibration parameters.

| Parameter | Description | File | Unit | Calibration inputs | | | Calibrated value |
|---|---|---|---|---|---|---|---|
| | | | | Initial value | Lower bound | Upper bound | |
| DEPIMP_BSN | Depth to impervious layer for modeling perched water tables. | .bsn | mm | 3000 | 0 | 6000 | 1356 |
| EPCO | Plant uptake compensation factor. | .bsn | - | 0.5 | 0.01 | 1 | 0.2306 |
| SFTMP | Snowfall temperature. | .bsn | °C | 0 | -5 | 5 | 1.381 |
| SMFMN | Minimum melt rate for snow during year. | .bsn | mm/°C-day | 5 | 0 | 10 | 2.078 |
| SMFMX | Maximum melt rate for snow during year. | .bsn | mm/°C-day | 5 | 0 | 10 | 2.078 |
| SMTMP | Snow melt base temperature. | .bsn | °C | 0 | -5 | 5 | -0.9346 |
| SNO50COV | Snow water content that corresponds to 50% snow cover. | .bsn | mm | 0.5 | 0.01 | 0.99 | 0.3092 |
| SNOCOVMX | Minimum snow water content that corresponds to 100% snow cover. | .bsn | mm | 1 | 1 | 650 | 152.1 |
| SURLAG | Surface runoff lag time. | .bsn | day | 4 | 1 | 24 | 12.5 |
| TIMP | Snow pack temperature lag factor. | .bsn | - | 0.5 | 0.01 | 1 | 0.5362 |
| ADJ_PKR | Peak rate adjustment factor for sediment routing in the sub-basin. | .bsn | - | 1.25 | 0.5 | 2 | 1.052 |
| PRF | Peak rate adjustment factor for sediment routing in the channel. | .bsn | - | 1 | 0 | 2 | 1.803 |
| ALPHA_BF | Baseflow alpha factor. | .gw | days | 0.048 | 0 | 1 | 0.6387 |
| GW_DELAY | Groundwater delay. | .gw | day | 250 | 0 | 500 | 472.1 |
| GW_REVAP | Groundwater "revap" coefficient. | .gw | - | 0.1 | 0.02 | 0.2 | 0.04354 |
| GW_SPYLD | Specific yield of the shallow aquifer.* | .gw | $m^3/m^3$ | 0.25 | -0.5 | 1 | -0.08856 |
| GWHT | Initial groundwater height. | .gw | m | 12.5 | 0 | 25 | 1.101 |
| GWQMN | Threshold depth of water in the shallow aquifer for return flow to occur. | .gw | mm | 2500 | 0 | 5000 | 4442 |
| RCHRG_DP | Deep aquifer percolation fraction. | .gw | - | 0.05 | 0 | 1 | 0.2275 |
| REVEP_MN | Threshold depth of water in the shallow aquifer for "revap" to occur. | .gw | mm | 250 | 0 | 500 | 472.9 |
| CANMX | Maximum canopy storage. | .hru | mm | 0 | 0 | 10 | 3.057 |
| ESCO | Soil evaporation compensation factor. | .hru | - | 0.05 | 0.01 | 1 | 0.3678 |
| OV_N | Manning's "n" value for overland flow. | .hru | - | 0.15 | 0.01 | 0.3 | 0.2764 |
| SLOPE | The mean slope within the HRU.* | .hru | m/m | 0 | -0.1 | 0.1 | -0.09433 |
| DEP_IMP | Depth to impervious layer in soil profile. | .hru | mm | 2000 | 1500 | 2500 | 2304 |
| SLSUBBSN | Average slope length. | .hru | m | 50 | 10 | 150 | 90.45 |
| DDRAIN | Depth to subsurface drain. | .mgt | mm | 1000 | 500 | 1500 | 1173 |
| TDRAIN | Time to drain soil to field capacity. | .mgt | hr | 36 | 0 | 72 | 55.54 |





Continued.

| Parameter | Description | File | Unit | Calibration inputs | | | Calibrated value |
|---|---|---|---|---|---|---|---|
| | | | | Initial value | Lower bound | Upper bound | |
| CH_KII | Effective hydraulic conductivity in main channel alluvium. | .rte | Mm/hr | 256 | -0.01 | 500 | 401.2 |
| CH_NII | Manning's "n" value for the main channel. | .rte | - | 0.15 | 0.01 | 0.3 | 0.0255 |
| CH_SII | Average slope of main channel* | .rte | m/m | 0 | -0.05 | 0.05 | 0.02677 |
| SOL_AWC | Available water capacity.* | .sol | mm/mm | 1 | -0.1 | 2 | 0.9813 |
| SOL_K | Saturated hydraulic conductivity.* | .sol | mm/hr | 2 | -0.5 | 5 | -0.4585 |
| SOL_ALB | Moist soil albedo.* | .sol | - | 0.25 | -0.5 | 1 | -0.3694 |
| SOL_Z | Depth from soil surface to bottom layer.* | .sol | mm | 0.25 | -0.5 | 1 | -0.1593 |
| CH_KI | Effective hydraulic conductivity in tributary channel alluvium. | .sub | mm/hr | 150 | 0 | 300 | 244.2 |
| CH_NI | Manning's "n" value for the tributary channels. | .sub | - | 0.15 | 0.008 | 0.3 | 0.2437 |
| CH_SI | Average slope of tributary channels.* | .sub | m/m | 0 | -0.05 | 0.05 | -0.02402 |

* These parameters were varied as a percentage of to maintain spatial variability.