# Peer review of "Assessing the hydrologic response to wildfires in mountainous regions"

_Hydrology and Earth System Sciences, 2017_

## Referee Comment (RC1) · Anonymous Referee #1 · 2 Dec 2017

General Comments:

Havel et al. present an assessment of hydrologic response to wildfire using the SWAT model, applied at multiple spatial scales. The authors provide a good overview of the problem and clearly state the goal and objectives. The paper title and primary goal, aimed at characterizing and quantifying long-term hydrologic responses to wildfires in mountainous regions are perhaps a little misleading. The study analyzed runoff (using SWAT) for the study domain over a period of 2000-2014, with the wildfires occurring in 2012. Therefore, the burned condition is only represented in the short-term. The title and all such references/inferences in the paper to long-term effects of fire should be revised to more clearly depict unburned and burned periods evaluated. The methodology is explained reasonably well, except where indicated below, and substantial material is

[Figure]

provided through appendices. The findings are supported reasonably well throughout except where noted below. The study is relevant within the scope of HESS, although the novelty needs better depiction. Conclusions could be more substantial. The length of the paper and associated elements is appropriate.

Specific comments: Abstract, Page 1, Line 8: The verbiage "long-term hydrologic responses to wildfires" seems misleading given only the first few years post-fire (2012-2014) are part of a longer-term analysis (2000-2014). This should be addressed here and throughout to more clearly state what was evaluated.

Abstract, Page 1: The abstract doesn't really present substantially novel findings. The results are somewhat typical for burned watersheds. Consider clearly presenting novel components of the work along with the primary findings.

Pages 2-3, Lines 30-32 and Lines 1-8: I'm not sure this content achieves the intended (assume to provide justification for the current study). The text here suggests the requisite approaches use static variables to represent dynamic properties. While this approach does have limitations, utilizing a dynamic approach that may not fully represent the dynamics at hand also has limitations. Anyway, the text doesn't necessary make a compelling case for one approach over another. Also, the word "components" need some reference/definition within this text.

Page 5, Lines 7-9: This is a very broad statement and assumes the continuous model is accurately depicting the dynamic events. Perhaps additional citations would better support this statement as the norm. I'm not entirely convinced the approach used in this study demonstrates a better representation or just a different one. Both approaches can be effective, useful, and yield good results. Any general commentary on one versus the other should be clearly justified and include substantial citations in support.

Pages 5-6, Lines 24-30 and Lines 1-4: More specifics needed here and explanation of Range-Grasses approach is merited.

Page 9, Line 2: I'm not sure I agree that the error statistics tell how accurately SWAT is representing processes exactly.

Page 12, Line 13: The inference that it may be reasonable to use total burn area percentage as a predictor requires some qualification here given the single study.

---

## Referee Comment (RC2) · Anonymous Referee #2 · 3 Dec 2017

Main remarks

The manuscript deals with an interesting theme for HESS, both for a hydrological point of view and for a soil science point of view. Namely the authors have tackled a very complex problem that is rising the international audience interest, the catchment hydrologic response to wildfires. The authors have developed calibrated and validated a numerical model using a well-known code SWAT to answer this question. As far as I can judge, the English of the manuscript is good and the grammar correct. The title of the manuscript is self-explicative. The abstract provides the necessary information to be considered stand alone. The keywords are all necessary and pertinent. The introduction section is pertinently review the processes of wildfire induce on hydrologic behavior of catchments, but is not actually appropriate in describing the mathematical

approaches. In fact, it lacks to review the state of the art of various modelling approaches that have been widely used in the last 20 years in hydrologic modelling, e.g. Clark et al. 2017 Hydrol. Earth Syst. Sci., 21, 3427–3440. And how this model can be considered among the variety of approaches so far used.

The materials and methods sections is very well constructed and I do appreciate the information available as appendix to fully explain the model construction and to allow repeatability of model set up. The model has been calibrated and validated versus flow data at the outlet of the catchment using 38 parameters, the number of parameters is quite elevated thus non-uniqueness of the solution is probable in such inverse problems. For this reason, the authors should at least provide more information about the sensitivity of model parameters. In fact, they claim that additional parameters respect to Foy at al. 2015 were introduced following the results of Ahmadi et al., 2014 (that conducted a study also on solute concentrations in another watershed), but no information is actually provided on which parameters were added and which global sensitivities they have found. In addition, since here only flow discharge is used as observation data, the authors should comment on the limitations of using a single source of data to calibrate a model. Finally, a key factor controlling the hydrological base flow is the parameterization of the saturated hydraulic conductivity field, that could be highly heterogeneous at the Hydrologic Soil Group scale, thus it would be useful to show sensitivities to this particular parameter.

Overall the results and discussion section should be improved with the above mentioned suggestions to better support the conclusions of this paper.

In my opinion the manuscript presents novel and robust data to evaluate the long-term hydrologic response to wildfires, so I feel that the paper could be accepted for publication in HESS after the minor corrections recommended.

Specific remarks

Figure 2 the terrain slope range is scale is too high (0-9999) and the map display only

one colour, please amend the figure.

Figure 2 the number "3" in the right upper corner is affected by distortion" please correct it.

Figure 2 The Watershed Outlet symbol is not visible in this figure and is only visible in fig.1, please correct it.

---

## Author Comment (AC1) · 25 Jan 2018

**Response to anonymous reviewer #1:**

The authors of the paper would like to thank the anonymous reviewer for their valuable and insightful comments. We tried to carefully address all of the comments from the reviewer. The details of the revision are included below. In the beginning of the file the direct response to the comments is presented and after that the details of the revision in the manuscript is added:

**The paper title and primary goal, aimed at characterizing and quantifying long-term hydrologic responses to wildfires in mountainous regions are perhaps a little misleading.**
**Abstract, Page 1, Line 8: The verbiage "long-term hydrologic responses to wildfires" seems misleading given only the first few years post-fire (2012-2014) are part of a longer-term analysis (2000-2014). This**
10  **should be addressed here and throughout to more clearly state what was evaluated.**

Regarding the title of the paper which contains long-term responses, the reason for using this term was that many studies are conducted on investigating the immediate response of the watersheds to wildfires (next few precipitation events after the fire). We used the term "long-term" to indicate that our study has been done to explore the
15  hydrologic effects of the fire on continuous-time basis for a few years after the fire in contrast to a few precipitation events. Anyway, we removed the term long-term from the title to avoid any misinterpretation. We explained what we meant by long-term in the abstract and throughout the manuscript and explicitly mentioned the two year period for analyzing the effects of the fire in the abstract and other locations in the manuscript.

20  **Abstract, Page 1: The abstract doesn't really present substantially novel findings. The results are somewhat typical for burned watersheds. Consider clearly presenting novel components of the work along with the primary findings.**

Thanks. The abstract was revised to emphasis more on the novelty of the approach and important findings and
25  implications.

**Pages 2-3, Lines 30-32 and Lines 1-8: I'm not sure this content achieves the intended (assume to provide justification for the current study). The text here suggests the requisite approaches use static variables to represent dynamic properties. While this approach does have limitations, utilizing a dynamic approach**
30  **that may not fully represent the dynamics at hand also has limitations. Anyway, the text doesn't necessary make a compelling case for one approach over another. Also, the word "components" need some reference/definition within this text.**

That is a valid point that lacking the ability to capture the full dynamic of the problem has limitations. However,
35  incorporating a dynamic approach even with its limitations can enhance the physical base of the simulation and help with more realistic representation of the processes compared to a static approach. We added more references and some more description of the method to better show the merits for its application. By component we mean the modules in the model that provide the capability to implement a specific change in model. Here, LULC component is the land use change module. The explanation is added to the manuscript.

**Page 5, Lines 7-9: This is a very broad statement and assumes the continuous model is accurately depicting the dynamic events. Perhaps additional citations would better support this statement as the norm. I'm not entirely convinced the approach used in this study demonstrates a better representation or just a different one. Both approaches can be effective, useful, and yield good results. Any general commentary on one versus the other should be clearly justified and include substantial citations in support.**

Both event-based and continuous-time models are useful and one can better represent processes compared to the other depending on the study problem. We did not intend to imply that continuous-time models are better than event-based models or they represent the processes accurately. For this study we had to use a continuous time model and in this paragraph we explain why a continuous time model better represents the processes involved in this specific problem. Some more references were added to the text for this argument.

**Pages 5-6, Lines 24-30 and Lines 1-4: More specifics needed here and explanation of Range-Grasses approach is merited.**

Thanks. The reason for updating the LULC to Range-Grasses was explained in more detail in the text.

**Page 9, Line 2: I'm not sure I agree that the error statistics tell how accurately SWAT is representing processes exactly.**

We used the error statistics to assess how good the model simulations conform to real observations. While the good error statistics does not guaranty that the hydrologic processes are represented completely accurate, it gives a good insight into the performance of the model. The text was reworded to avoid confusion.

**Page 12, Line 13: The inference that it may be reasonable to use total burn area percentage as a predictor requires some qualification here given the single study.**

The idea that increase in percent burn area results in increase in runoff has been reported in several studies (Benavides-Solorio and MacDonald, 2001; Inbar et al., 1998; Lavabre et al., 1993; Robichaud et al., 2000; Scott, 1993). However, we were not able to identify studies where the relationship between runoff increase and percent total burned area was quantified similar to our study. Given the regression analysis done in this study and checking the metrics indicating the strength and significance of the regression model, we expressed that percentage of total burned area may be useful as an indicator of increase in runoff. Anyway to be more accurate we added "in the Cache la Poudre watershed" since as you have mentioned this is a single study on a single watershed. We also mentioned that more studies are needed to generalize the findings of the study.

[revised manuscript text omitted]

---

## Author Comment (AC2) · 25 Jan 2018

**Response to anonymous reviewer #2:**

The authors of the paper would like to thank the anonymous reviewer for their valuable and insightful comments. We tried to carefully address all of the comments from the reviewer. The details of the revision are included below. In the beginning of the file the direct response to the comments is presented and after that the details of the revision in the manuscript is added:

**The introduction section is pertinently review the processes of wildfire induce on hydrologic behavior of catchments, but is not actually appropriate in describing the mathematical approaches. In fact, it lacks to review the state of the art of various modelling approaches that have been widely used in the last 20 years in hydrologic modelling, e.g. Clark et al. 2017 Hydrol. Earth Syst. Sci., 21, 3427–3440. And how this**
10 **model can be considered among the variety of approaches so far used.**

> Thanks for the reminder about the introduction with regard to comparing our approach with the literature. We added some sentences and relevant references to the introduction to address this.

15 **The authors should at least provide more information about the sensitivity of model parameters. In fact, they claim that additional parameters respect to Foy at al. 2015 were introduced following the results of Ahmadi et al., 2014 (that conducted a study also on solute concentrations in another watershed), but no information is actually provided on which parameters were added and which global sensitivities they have found.**

> With regard to the sensitivity analysis, table A6 is showing the parameters from the sensitivity analysis in the work by Sandhya et al., 2014, Ahmadi et al., 2014; and Foy et al., 2015 (All done in our research group). The eFAST method was used by Sandhya et al. (2014) to derive the sensitivity indices and determine the parameters that model simulations show the highest sensitivity to. 30 parameters were selected based on that study (the study was done on
25 the same watershed as our study). Ahmadi et al. (2014) used the method of Sobol to identify the most sensitive parameters simulating streamflow and nutrients in a different watershed. Based on both studies, 38 parameters were selected for calibration. Table A6 in the appendices shows the selected parameters based on these sensitivity analysis studies for the watershed. Relevant description of the studies were added to the manuscript.

30 **Since here only flow discharge is used as observation data, the authors should comment on the limitations of using a single source of data to calibrate a model**

> Thanks. We discussed limitations of using a single source of data for calibrating the model.

35 **A key factor controlling the hydrological base flow is the parameterization of the saturated hydraulic conductivity field, that could be highly heterogeneous at the Hydrologic Soil Group scale, thus it would be useful to show sensitivities to this particular parameter.**

> We added some sentences in the results section explaining the sensitivity of model simulations to saturated
40 hydraulic conductivity.

**Figure 2 the terrain slope range is scale is too high (0-9999) and the map display only one colour, please amend the figure.**

Thank you. The map was revised to include multiple classes of slope.

**Figure 2 the number "3" in the right upper corner is affected by distortion" please correct it.**

We were not able to see where that number "3" is that was referred to.

10 **Figure 2 The Watershed Outlet symbol is not visible in this figure and is only visible in fig.1, please correct it.**

Thank you. We changed the symbol to make it more visible.

[revised manuscript text omitted]

---

## Author Response (AR2)

**Response to the Editor:**

The authors would like to thank the Editor and reviewers again for their valuable and insightful comments.
* * *
**Editor Decision: Publish subject to minor revisions (review by editor)** (13 Mar 2018) by Thom Bogaard

5   Comments to the Author:

Dear authors

you improved the paper considerably and I think publication is well justified. However, the reviewer makes a strong case whether you can make statements on long-term effects in hydrology. I support the reviewers request to be (case) specific and revising the text where the term long-term is used.

10   I look forward receiving the revised text in due time
* * *
**Response:**

In response to editor's comment requesting to revise the manuscript wherever the term long-term is used, we removed the term from the manuscript and explicitly explained the two-year period after the wildfire for which the study was conducted.

15   We carefully reviewed the manuscript one more time to make sure there are no typos or any other issues.

[revised manuscript text omitted]